# Estimated glomerular filtration rate equations in people of self-reported black ethnicity in the United Kingdom: Inappropriate adjustment for ethnicity may lead to reduced access to care

Rouvick M. Gama[1]ᴼ, Amanda Clery[2]ᴼ, Kathryn Griffiths[1], Neil Heraghty[3], Adrien M. Peters[3], Kieran Palmer[1], Henry Kibble[1], Royce P. Vincent[4], Claire C. Sharpe[5], Hugh Cairns[1], Kate Bramham[1,6]*

1 King's Kidney Care, King's College Hospital NHS Foundation Trust, London, United Kingdom, 2 School of Population Health and Environmental Sciences, King's College London, London, United Kingdom, 3 Department of Nuclear Medicine, King's College Hospital NHS Foundation Trust, London, United Kingdom, 4 Department of Clinical Biochemistry, King's College Hospital NHS Foundation Trust, London, United Kingdom, 5 Department of Inflammation Biology, School of Immunology and Microbial Sciences, King's College London, London, United Kingdom, 6 Department of Women and Children's Health, King's College London, London, United Kingdom

ᴼ These authors contributed equally to this work.
* kate.bramham@kcl.ac.uk

**Data Availability Statement:** All relevant data are within the manuscript.

## Abstract

Assessment in African populations suggest adjustment for ethnicity in estimated glomerular filtration rate (eGFR) equations derived from African Americans lead to overestimation of GFR and failure to determine severity in chronic kidney disease (CKD). However, studies in African Europeans are limited. We aimed to assess accuracy of eGFR equations, with and without ethnicity adjustment compared with measured GFR in people of Black ethnicity in the United Kingdom. Performance of MDRD, CKD-EPI (with and without ethnicity adjustment), Full Age Spectrum (FAS), revised Lund Malmö (LM Revised), and European Kidney Function Consortium (EKFC) eGFR equations were assessed compared to $^{51}$Cr-EDTA GFR studies extracted from hospital databases. Participants with albumin <30g/l, liver disease, <18 years, of non-Black or non-White self-reported ethnicity were excluded. Agreement was assessed by bias, precision and 30%-accuracy and was stratified for ethnicity and GFR. 1888 $^{51}$Cr-EDTA studies were included (Mean age-53.7yrs; 43.6% female; 14.1% Black ethnicity). Compared to White participants, eGFR-MDRD and eGFR-CKD-EPI equations in Black participants significantly overestimated GFR (bias 20.3 and 19.7 ml/min/1.73m² respectively, p<0.001). Disregarding the ethnicity adjustment significantly improved GFR estimates for Black participants (bias 6.7 and 2.4ml/min/1.73m² for eGFR-MDRD and eGFR-CKD-EPI respectively, p<0.001). The LM Revised equation had the smallest bias for both White and Black participants (5.8ml and -1.1ml/min/1.73m² respectively). 30%-accuracy was superior for GFR≥60ml/min/1.73m² compared to <60ml/min/1.73m² using eGFR-CKD-EPI equation for both White and Black participants (p<0.001). Multivariate regression

**Funding:** The authors received no specific funding for this work.

**Competing interests:** The authors have declared that no competing interests exist.

methodology with adjustment for age, sex and log(serum creatinine) in the cohort yielded an ethnicity coefficient of 1.018 (95% CI: 1.009–1.027). Overestimation of measured GFR with eGFR equations using ethnicity adjustment factors may lead to reduced CKD diagnosis and under-recognition of severity in people of Black ethnicity. Our findings suggest that ethnicity adjustment for GFR estimation in non-African Americans may not be appropriate for use in people of Black ethnicity in the UK.

## Introduction

Chronic kidney disease (CKD) is a global health problem, with adverse outcomes including end stage kidney disease (ESKD), cardiovascular disease and premature death [1,2]. Gold standard assessment of Glomerular Filtration Rate (GFR) with formal methods is costly, impractical, and not readily available, thus estimated GFR (eGFR) is commonly used in clinical practice. The most widely used equations for estimating GFR are the four variable Modified Diet in Renal Disease (MDRD) and chronic kidney disease epidemiology collaboration (CKD-EPI) equations [3–5]. Both equations, which were derived from cohorts including African Americans, recommend use of an adjustment factor for Black ethnicity (CKD-EPI 1.159; MDRD 1.212) to enhance accuracy [6,7].

Several studies have reported that there are higher rates of production of creatinine in people of Black ethnicity [8–12], which support this practice, and it has been assumed that eGFR adjustment should apply to all of Black ethnicity. However, recent studies from different African countries suggest eGFR equations more accurately reflect measured GFR (mGFR) without ethnicity adjustment [5,6,13–18], but studies of eGFR equations in African Europeans are scarce. Even in the USA, several bodies are recognising the limitations of 'racial categorisation', including people with mixed ancestry [19].

People of African or Afro-Caribbean ancestry are at greater risk of CKD [20,21], have more rapid progression of disease, a higher incidence of ESKD and have more advanced disease at presentation compared to Caucasians. Thus accurate assessment of GFR is important for early diagnosis, risk stratification and timely management [22,23]. Early diagnosis of CKD is critical in order to implement preventative strategies, particularly in low-income countries where prevalence of CKD due to non-communicable and infectious diseases is estimated to be high and renal replacement therapy (RRT) may not be easily accessible [6,17,18,24,25]. Inappropriate use of ethnicity adjusted eGFR equations could further contribute to recognised ethnicity related health inequalities in CKD due to delayed diagnosis, preparation for renal replacement therapy and wait-listing for transplantation. To address this, research has been recommended to quantify the benefits and harms of using race in GFR estimation [26], and use of alternative measures of GFR [27].

This study aimed to assess the accuracy of eGFR equations, (MDRD and CKD EPI with and without ethnicity adjustment, Full Age Spectrum (FAS), revised Lund Malmö (LM Revised), and European Kidney Function Consortium (EKFC)) compared with gold standard chromium-51 labelled ethylenediaminetetraacetic acid ($^{51}$Cr-EDTA) clearance assays and the impact of eGFR assessment on clinical care in a large population in the United Kingdom.

## Methods

We conducted a single-centre observational cross-sectional study in a large tertiary hospital in London, United Kingdom. All $^{51}$Cr-EDTA studies were extracted between 2009–2019 from hospital databases: Laboratory Information Management System (Clinisys) and Sunrise

Electronic Patient Records (EPR). Baseline characteristics, including age, gender, self-reported ethnicity, referral specialty, number of [51]Cr-EDTA studies and serum creatinine (IDMS traceable assay) and albumin concentrations taken within one week of [51]Cr-EDTA study were recorded. For individuals with repeated mGFR assessments, only the first mGFR assessment was included in the analysis.

Exclusions were made if: (1) creatinine or albumin measurements were taken more than a week from the [51]Cr-EDTA study, (2) albumin measurements were <30g/L (due to potentially reduced muscle mass), (3) referrals were from liver or rehabilitation services, due to liver disease interference with creatinine levels and likely amputation respectively, (4) patients were under 18 years old, (5) self-reported ethnicity was non-Black or non-White, or mixed ethnicities, and (6) there were any incomplete data in the previous criteria.

Serum creatinine and albumin concentrations were determined using the Siemens clinical chemistry analysers (Advia 2400, Siemens Diagnostics, Frimley, UK) in an UK Accreditation Service (UKAS) accredited laboratory. Creatinine assay was the Jaffe method with an isotope dilution-mass spectrometry (IDMS) traceable calibrator.

mGFR was measured by [51]Cr-EDTA, administered intravenously (10 MBq). Plasma clearance of the tracer was calculated from accurately-timed plasma samples obtained at 120, 180 and 240 min, and corrected for the assumption of a single compartment using the formula of Bröchner-Mortensen [28].

eGFRs were calculated using the following equations: Chronic Kidney Disease Epidemiology Collaboration (CKD-EPI), with and without an ethnicity correction factor, Modification of Diet in Renal Disease (MDRD), with and without an ethnicity correction factor, Full Age Spectrum (FAS), revised Lund Malmö (LM Revised), and the European Kidney Function Consortium (EKFC; Table 1).

## Statistical methods

All demographic characteristics (age, sex, BMI, referral source and CKD stage) were summarised using counts and percentages for categorical variables and mean with standard deviation

**Table 1. Estimated glomerular filtration rate (eGFR) equations.**

| | |
|---|---|
| CKD-EPI adjusted | $141 \times \min(SCr/\kappa, 1)^{\alpha} \times \max(SCr/\kappa, 1)^{-1.209} \times 0.993^{Age} \times 1.018[\text{if female}] \times 1.159[\text{if Black}]$<br>$\alpha = -0.329[\text{if female}]$ or $-0.411[\text{if male}]$<br>$\kappa = 0.7[\text{if female}]$ or $0.9[\text{if male}]$<br>min and max indicate the minimum or maximum of $SCr/\kappa$ or 1, respectively |
| CKD-EPI unadjusted | $141 \times \min(SCr/\kappa, 1)^{\alpha} \times \max(SCr/\kappa, 1)^{-1.209} \times 0.993^{Age} \times 1.018[\text{if female}]$ |
| MDRD adjusted | $175 \times SCr^{-1.154} \times Age^{0.203} \times 0.742[\text{if female}] \times 1.212[\text{if Black}]$ |
| MDRD unadjusted | $1.75 \times SCr^{-1.154} \times Age^{0.203} \times 0.742[\text{if female}]$ |
| FAS [29] | $107.3/(SCr/Q)$ [if aged $2 \leq 40$yr]<br>$107.3/(SCr/Q) \times 0.988^{Age-40}$ [if age>40yr]<br>Q = median SCr value for age-/sex-specific healthy populations |
| LM Revised [30] | $e^{X-0.0158 \times Age + 0.438 \times \ln(Age)}$<br>$X = 2.50 + 0.0121 \times (150-SCr)[\text{if female and } SCr<150]$<br>$X = 2.50 - 0.926 \times \ln(SCr/150)[\text{if female and } SCr \geq 150]$<br>$X = 2.56 + 0.00968 \times (180-SCr)[\text{if male and } SCr<180]$<br>$X = 2.56 - 0.926 \times \ln(SCr/180)$ [if male and $SCr \geq 180$]<br>$SCr = \mu mol/L$ |
| EKFC [31] | $107.3 \times (SCr/Q)^{-0.322}$ [if aged $2 \leq 40$yr and $(SCr/Q)<1$]<br>$107.3 \times (SCr/Q)^{-1.132}$ [if aged $2 \leq 40$yr and $(SCr/Q) \geq 1$]<br>$107.3 \times (SCr/Q)^{-0.322} \times 0.990^{Age-40}$ [if aged>40yr and $(SCr/Q<1$]<br>$107.3 \times (SCr/Q)^{-1.132} \times 0.990^{Age-40}$ [if aged>40yr and $(SCr/Q) \geq 1$]<br>Q values described in detail elsewhere. |

for continuous variables (or median and interquartile range for non-normal continuous variables). Characteristics were summarised for the whole cohort as well as stratified by ethnicity. Agreement was tested between mGFR and adjusted and unadjusted eGFR-MDRD and eGFR-CKD-EPI. Agreement was assessed using bias (mean difference between eGFR and mGFR), precision (SD of the bias), limits of agreement (bias +- 2 times precision), and 30% accuracy (proportion of eGFRs that were within 30% of mGFR value). Agreement was also assessed by ethnicity (White and Black) and then by GFR ($<$60 and $\geq$60 mL/min/1.73m$^2$).

Post-hoc analyses were conducted to explore agreement further. An ethnicity correction factor in our sample was calculated using multivariable regression analysis between ethnicity and mGFR, adjusting for age, sex and log (SCr). The association between serum creatinine and ethnicity was explored, as well as reasons for high bias between mGFR and eGFR using multivariable linear regression analyses.

To determine the impact of using ethnicity correction factors on current clinical care, the most recent creatinine for all participants of self-reported Black ethnicity attending nephrology services was extracted from the hospital database EPR. Deceased patients and those receiving haemodialysis or peritoneal dialysis were excluded. The number of patients with each CKD stage was calculated according to adjusted and unadjusted eGFR, albuminuria. The proportion of patients categorized by CKD Stage using adjusted and unadjusted eGFR-CKD-EPI and with eGFR $<$20 mL/min/1.73m$^2$ who would be referred for RRT planning according to local policy are summarised using counts and percentages. P-values $<$0.05 was considered to be significant. Data were analysed using statistical software R, version 3.6.

Data were extracted from King's College Hospital NHS Foundation Trust, London laboratory databases between Jan 2009 to Dec 2019. All data were fully anonymised prior to access by the researchers. The study was reviewed locally and was not considered to need research ethics committee approval and was registered on the King's College Hospital NHS Foundation Trust Nephrology Audit Register 2019 (KCH/KKC/2020:003).

## Results

After exclusion of participants with predefined confounders ((1) creatinine or albumin measurements were taken more than a week from the $^{51}$Cr-EDTA study, (2) albumin measurements were $<$30g/L (due to potentially reduced muscle mass), (3) referrals were from liver or rehabilitation services, due to liver disease interference with creatinine levels and likely amputation respectively, (4) patients were under 18 years old, (5) self-reported ethnicity was non-Black or non-White, or mixed ethnicities) which might influence serum creatinine concentration, a total of 1888 $^{51}$Cr-EDTA mGFR studies were identified (266 (14.1%) Black; 1622 White) (Fig 1). Mean age of participants at the time of mGFR test was 53.7 years, 43.6% were female, mean BMI was 26.9 kg/m$^2$, and haematology was the most common referral source (71.7% of patients, Table 2).

Serum creatinine values were positively skewed. Median values were 71 μmol/L for White patients (IQR: 60–85) and 76 μmol/L (IQR: 64–101) for Black patients, and tended to be higher for Black men than White men. The mean GFR, measured using $^{51}$Cr-EDTA, was 77ml/min/1.73m$^2$ and ranged from 4-162ml/min/1.73m$^2$ (Fig 2). This was similar for both Black and White patients (Table 2). The eGFR equation with the highest mean was the MDRD using the ethnicity correction factor, 92.3ml/min/1.73m$^2$ and the lowest mean was the LM Revised equation, 81.8ml/min/1.73m$^2$.

All eGFR equations tended to overestimate mGFR and the biases fell across an approximately normally distributed range of measurements (Fig 3). For all 1622 White participants, the eGFR FAS equation had the greatest bias of 15.2ml/min/1.73m$^2$ followed by MDRD

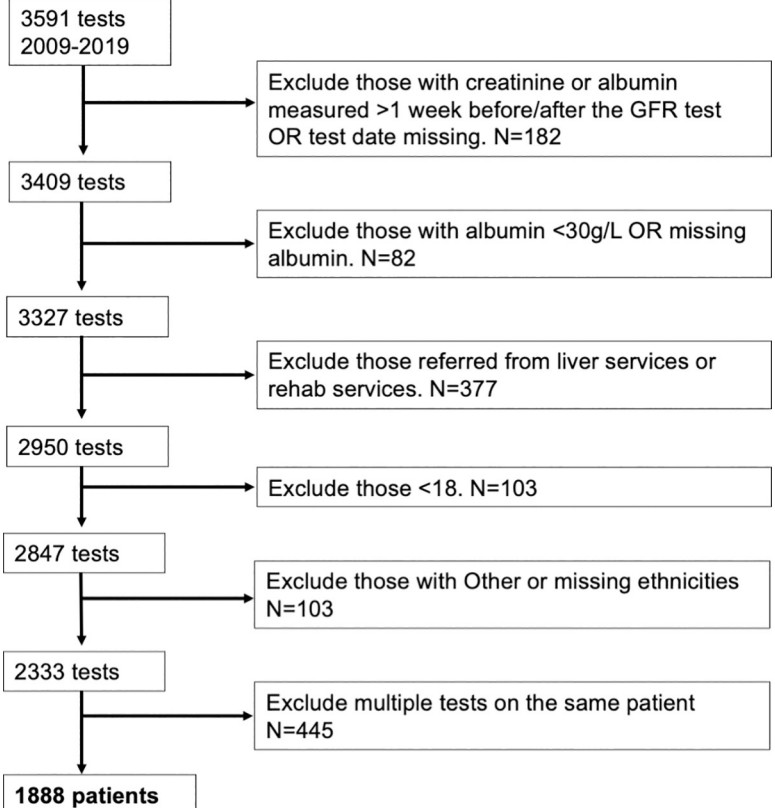

**Fig 1. Flowchart of exclusion criteria.**

equation with a bias of 14.6ml/min/1.73m$^2$ (Table 3). The LM Revised equation had the smallest bias, 5.8ml/min/1.73m$^2$. In all 266 Black participants, the CKD-EPI equation adjusted with the ethnicity correction factor had the highest bias (20.3ml/min/1.73m$^2$) and the LM Revised had the smallest (-1.1ml/min/1.73m$^2$). Discounting the ethnicity correction factor, the bias of both CKD-EPI and MDRD equations was significantly reduced (6.7 and 2.4ml/min/1.73m$^2$ respectively, p<0.001). This also improved the 30% accuracy of equations for Black participants: from 56.4% to 77.1% (p<0.001) for CKD-EPI and from 56.8% to 75.2% (P<0.001) for MDRD equations.

All equations had higher 30% accuracy for both Black and White participants in those with an mGFR≥60ml/min/1.73m$^2$ compared to <60ml/min/1.73m$^2$. In the 48 Black participants with mGFR<60ml/min/1.73m$^2$, the LM Revised equation provided the highest 30% accuracy of 66.7%. This equation, along with the EKFC, provided the highest 30% in the 218 Black participants with mGFR≥60ml/min/1.73m$^2$ at 84.9%. In White participants, the LM Revised equation also provided the highest 30% accuracy of 55.0% and 89.8% in the <60 and ≥60ml/min/1.73m$^2$ groups, respectively (Table 3).

All equations had higher 30% accuracy for both Black and White participants in those with an mGFR≥60ml/min/1.73m$^2$ compared to <60ml/min/1.73m$^2$. In the 48 Black participants with mGFR<60ml/min/1.73m$^2$, the LM Revised equation provided the highest 30% accuracy of 66.7%. This equation, along with the EKFC, provided the highest 30% in the 218 Black participants with mGFR≥60ml/min/1.73m$^2$ at 84.9%. In White participants, the LM Revised equation also provided the highest 30% accuracy of 55.0% and 89.8% in the <60 and ≥60ml/min/1.73m$^2$ groups, respectively (Table 3).

**Table 2. Cohort characteristics, stratified by ethnicity.**

|  | All (N = 1888) | Black (N = 266) | White (N = 1622) |
|---|---|---|---|
| Age in years, mean (SD) | 53.7 (12.7) | 47.3 (11.5) | 54.7 (12.6) |
| Male, N (%) | 1065 (56.4) | 141 (53.0) | 924 (57.0) |
| Body Mass Index in kg/m², mean (SD) | 26.9 (5.3) | 27.0 (5.5) | 26.9 (5.3) |
| Referral source, N (%) |  |  |  |
| Haematology | 1353 (71.7) | 188 (70.7) | 1165 (71.8) |
| Oncology | 35 (1.9) | 8 (3.0) | 27 (1.7) |
| Renal | 192 (10.2) | 19 (7.1) | 173 (10.7) |
| Urology | 27 (1.4) | 4 (1.5) | 23 (1.4) |
| ICU | 2 (0.1) | 0 (0.0) | 2 (0.1) |
| Medicine | 115 (6.1) | 14 (5.3) | 101 (6.2) |
| Paediatrics* | 65 (3.4) | 12 (4.5) | 53 (3.3) |
| Other | 54 (2.9) | 5 (1.9) | 49 (3.0) |
| Unknown | 45 (2.4) | 16 (6.0) | 29 (1.8) |
| SCr in µmol/L, median (IQR) | 72 (60–86) | 76 (64–101) | 71 (60–85) |
| Male | 79 (68–93) | 93 (75–111) | 78 (67–90) |
| Female | 63 (53–74) | 65 (56–77) | 63 (53–73) |
| mGFR stage, N (%) |  |  |  |
| > = 90 | 518 (27.4) | 86 (32.3) | 432 (26.6) |
| 60–89 | 1024 (54.2) | 132 (49.6) | 892 (55.0) |
| 30–59 | 304 (16.1) | 36 (13.5) | 268 (16.5) |
| <30 | 42 (2.2) | 12 (4.5) | 30 (1.8) |
| GFR in ml/min/1.73m², mean (SD) |  |  |  |
| mGFR | 77.0 (21.3) | 78.9 (24.1) | 76.7 (20.8) |
| eGFR CKD-EPI adjusted | 92.1 (23.1) | 99.3 (30.6) | 91.0 (21.4) |
| eGFR CKD-EPI unadjusted | 90.2 (22.3) | 85.7 (26.4) | 91.0 (21.4) |
| eGFR MDRD adjusted | 92.3 (29.7) | 98.6 (35.8) | 91.2 (28.5) |
| eGFR MDRD unadjusted | 89.8 (28.8) | 81.3 (29.6) | 91.2 (28.5) |
| eGFR FAS | 91.2 (27.7) | 87.1 (29.1) | 91.8 (27.5) |
| eGFR LMRev | 81.8 (19.8) | 77.8 (23.0) | 82.5 (19.2) |
| eGFR EKFC | 85.7 (20.4) | 82.6 (24.0) | 86.2 (19.7) |

*potential organ donors to paediatric patients who were 18 years or older.

For all 1622 White participants, the eGFR FAS equation had the greatest bias of 15.2ml/min/1.73m² followed by MDRD equation with a bias of 14.6ml/min/1.73m² (Table 3). The LM Revised equation had the smallest bias, 5.8ml/min/1.73m². In all 266 Black participants, the CKD-EPI equation, adjusted with the ethnicity correction factor, had the highest bias (20.3ml/min/1.73m²) and the LM Revised had the smallest bias (-1.1ml/min/1.73m²). Without ethnicity correction factor, bias of both CKD-EPI and MDRD equations was significantly reduced (6.7 and 2.4ml/min/1.73m² respectively, p<0.001). Removal of ethnicity correction factor also improved the 30% accuracy of equations for Black participants: from 56.4% to 77.1% (p<0.001) for CKD-EPI and from 56.8% to 75.2% (P<0.001) for MDRD equations.

Bland-Altman plots showed that for most eGFR equations, the majority of participants fell within 2 standard deviations of the bias, which were evenly distributed (Fig 4). An exception is seen for both adjusted and unadjusted MDRD equations, as well as the FAS equation, where the bias tends to be greater at higher mean GFR values.

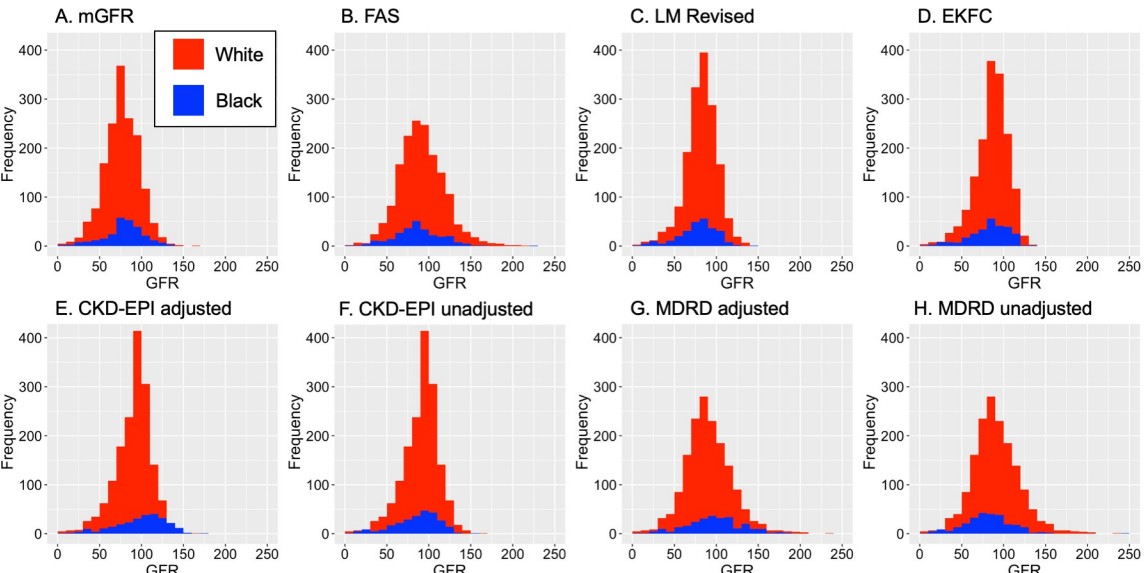

**Fig 2. Density plots of GFR measurements, stratified by ethnicity.** (**A**) mGFR, (**B**) eGFR FAS equation, (**C**) eGFR LM Revised equation, (**D**) eGFR EKFC equation, (**E**) eGFR CKD-EPI adjusted for ethnicity correction factor, (**F**) eGFR CKD-EPI unadjusted for ethnicity, (**G**) eGFR MDRD adjusted, and (**H**) eGFR MDRD unadjusted.

To explore agreement further, an ethnicity coefficient for our study sample was calculated, which was 1.018 (95% CI: 1.009–1.027; Table 4) after adjustment for age, sex and log(SCr). The association between ethnicity and serum creatinine was also explored. Black participants had a 13.8% higher (95% CI: 10.3–17.3%) SCr value compared to White participants, after adjustment for mGFR, age and sex (Table 5).

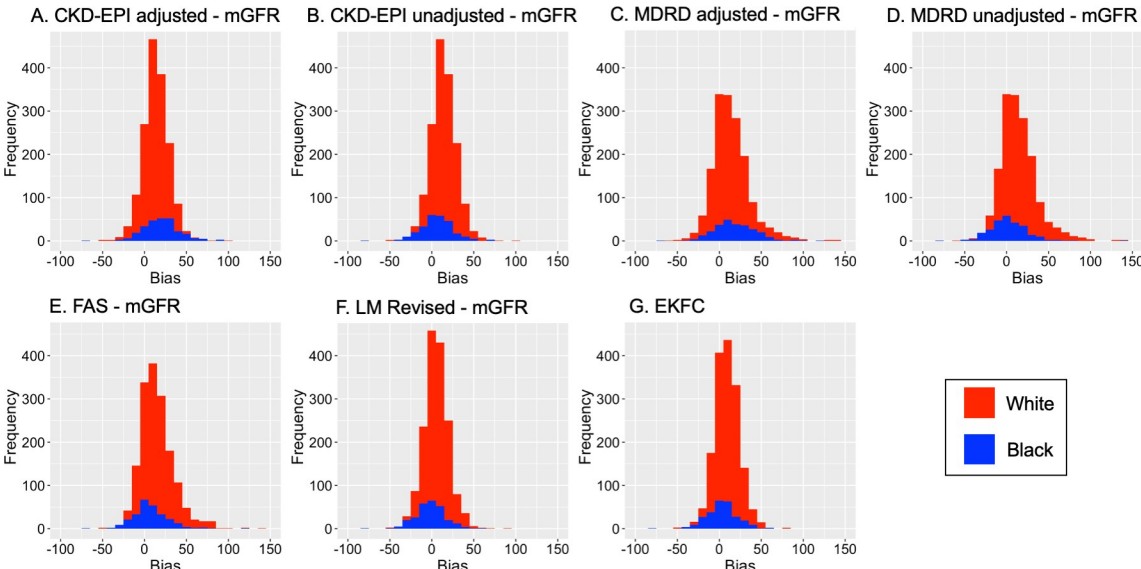

**Fig 3. Density plots of the bias between each eGFR equation and mGFR, stratified by ethnicity.** (**A**) eGFR CKD-EPI adjusted for ethnicity correction factor, (**B**) eGFR CKD-EPI unadjusted for ethnicity, (**C**) eGFR MDRD adjusted, (**D**) eGFR MDRD unadjusted, (**E**) eGFR FAS equation, (**F**) eGFR LM Revised equation, (**G**) eGFR EKFC equation.

**Table 3. Estimated glomerular filtration rate (eGFR) equations bias, precision and accuracy compared with mGFR, stratified by ethnicity.**

| | | | GFR, ml/min/1.73m² Mean (SD) | Median bias, ml/min/1.73m² | Mean bias, ml/min/1.73m² | Mean percentage bias, % | Precision, ml/min/1.73m² | Limits of agreement, ml/min/1.73m² | 30% Accuracy (%) |
|---|---|---|---|---|---|---|---|---|---|
| Black | All (N = 266) | mGFR vs: | 78.9 (24.1) | | | | | | |
| | | Adj CKD-EPI | 99.3 (30.6) | 20.0 | 20.3 | 29.5 | 21.8 | -23.3 to 63.9 | 56.4 |
| | | Unadj CKD-EPI | 85.7 (26.4) | 7.0 | 6.7 | 11.8 | 19.4 | -32.1 to 45.5 | 77.1 |
| | | Adj MDRD | 98.6 (35.8) | 16.0 | 19.7 | 28.1 | 27.1 | -34.5 to 73.9 | 56.8 |
| | | Unadj MDRD | 81.3 (29.6) | 1.0 | 2.4 | 5.6 | 22.8 | -43.2 to 48 | 75.2 |
| | | FAS | 87.1 (29.1) | 5.5 | 8.2 | 14.1 | 21.7 | -35.2 to 51.6 | 76.3 |
| | | LM Revised | 77.8 (23.0) | -1.0 | -1.1 | 2.0 | 17.6 | -36.3 to 34.1 | 81.6 |
| | | EKFC | 82.6 (24.0) | 4.0 | 3.7 | 8.5 | 18.1 | -32.5 to 39.9 | 80.5 |
| | mGFR <60 ml/min/1.73m² (N = 48) | mGFR vs: | 40.1 (13.9) | | | | | | |
| | | Adj CKD-EPI | 57.2 (27.6) | 12.0 | 17.1 | 44.1 | 20.4 | -23.7 to 57.9 | 43.8 |
| | | Unadj CKD-EPI | 49.4 (23.8) | 5.0 | 9.2 | 24.4 | 17.3 | -25.4 to 43.8 | 58.3 |
| | | Adj MDRD | 55.4 (25.4) | 10.5 | 15.2 | 40.5 | 18.3 | -21.4 to 51.8 | 50.0 |
| | | Unadj MDRD | 45.6 (20.9) | 2.5 | 5.5 | 15.8 | 14.7 | -23.9 to 34.9 | 64.6 |
| | | FAS | 51.1 (20.9) | 8.5 | 11.0 | 32.7 | 15.0 | -19 to 41 | 58.3 |
| | | LM Revised | 46.1 (21.0) | 3.5 | 6.0 | 17.5 | 14.6 | -23.2 to 35.2 | 66.7 |
| | | EKFC | 49.1 (22.2) | 4.5 | 9.0 | 25.3 | 16.0 | -23 to 41 | 60.4 |
| | mGFR > = 60 ml/min/1.73m² (N = 218) | mGFR vs: | 87.5 (16.1) | | | | | | |
| | | Adj CKD-EPI | 108.5 (22.4) | 21.0 | 21.1 | 26.3 | 22.0 | -22.9 to 65.1 | 59.2 |
| | | Unadj CKD-EPI | 93.7 (19.3) | 7.0 | 6.2 | 9.0 | 19.9 | -33.6 to 46 | 81.2 |
| | | Adj MDRD | 108.1 (30.4) | 17.5 | 20.7 | 25.3 | 28.6 | -36.5 to 77.9 | 58.3 |
| | | Unadj MDRD | 89.2 (25.1) | 0.5 | 1.7 | 3.4 | 24.2 | -46.7 to 50.1 | 77.5 |
| | | FAS | 95.0 (24.3) | 5.0 | 7.5 | 10.0 | 22.9 | -38.3 to 53.3 | 80.3 |
| | | LM Revised | 84.8 (16.7) | -2.5 | -2.7 | -1.3 | 17.9 | -38.5 to 33.1 | 84.9 |
| | | EKFC | 90.0 (17.2) | 4.0 | 2.5 | 4.8 | 18.3 | -34.1 to 39.1 | 84.9 |
| White | All (N = 1622) | mGFR vs: | 76.7 (20.8) | | | | | | |
| | | CKD-EPI | 91.0 (21.4) | 14.0 | 14.3 | 22.3 | 15.1 | -15.9 to 44.5 | 68.7 |
| | | MDRD | 91.2 (28.5) | 12.0 | 14.6 | 21.9 | 21.9 | -29.2 to 58.4 | 66.0 |
| | | FAS | 91.8 (27.5) | 13.0 | 15.2 | 22.8 | 19.8 | -24.4 to 54.8 | 67.7 |
| | | LM Revised | 82.5 (19.2) | 5.0 | 5.8 | 11.3 | 14.7 | -23.6 to 35.2 | 83.4 |
| | | EKFC | 86.2 (19.7) | 9.0 | 9.5 | 16.1 | 14.5 | -19.5 to 38.5 | 78.6 |
| | mGFR <60 ml/min/1.73m² (N = 298) | Corrected GFR vs: | 46.0 (11.9) | | | | | | |
| | | CKD-EPI | 63.7 (22.0) | 15.5 | 17.7 | 41.3 | 16.4 | -15.1 to 50.5 | 43.3 |
| | | MDRD | 60.9 (22.4) | 12.0 | 14.9 | 35.8 | 17.8 | -20.7 to 50.5 | 50.0 |
| | | FAS | 61.8 (20.6) | 13.0 | 15.8 | 39.3 | 16.2 | -16.6 to 48.2 | 51.7 |
| | | LM Revised | 58.9 (19.4) | 12.0 | 12.9 | 31.3 | 14.3 | -15.7 to 41.5 | 55.0 |
| | | EKFC | 61.1 (20.2) | 13.0 | 15.1 | 36.1 | 14.9 | -14.7 to 44.9 | 50.3 |
| | mGFR > = 60 ml/min/1.73m² (N = 1324) | Corrected GFR vs: | 83.6 (15.5) | | | | | | |
| | | CKD-EPI | 97.1 (15.8) | 14.0 | 13.5 | 18.1 | 14.7 | -15.9 to 42.9 | 74.5 |
| | | MDRD | 98.1 (25.1) | 12.0 | 14.5 | 18.8 | 22.7 | -30.9 to 59.9 | 69.6 |
| | | FAS | 98.6 (24.1) | 13.0 | 15.0 | 19.0 | 20.5 | -26 to 56 | 71.3 |
| | | LM Revised | 87.8 (14.6) | 4.0 | 4.2 | 6.8 | 14.4 | -24.6 to 33 | 89.8 |
| | | EKFC | 91.8 (14.5) | 8.0 | 8.2 | 11.7 | 14.1 | -20 to 36.4 | 85.0 |

mGFR = ⁵¹Cr-EDTA GFR: chromium-51 labelled ethylenediamine tetraacetic acid glomerular filtration rate corrected for body surface area; adj = adjusted with ethnicity correction factor; unadj = unadjusted with ethnicity correction factor.

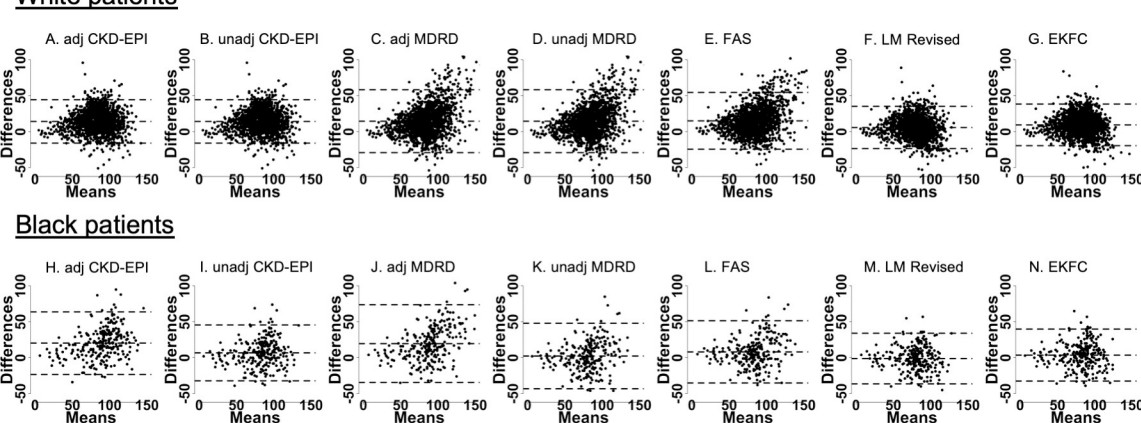

**Fig 4. Bland-Altman plots for eGFR equations compared to mGFR, stratified by ethnicity.** (White, Black) participants: **(A, H)** adjusted CKD-EPI, **(B, I)** unadjusted CKD-EPI, **(C, J)** adjusted MDRD, **(D, K)** unadjusted MDRD, **(E, L)** FAS, **(F, M)** LM Revised, **(G, N)** EKFC. x-axes show the mean between the mGFR and eGFR measurements ([mGFR+eGFR]/2) and y-axes show the differences between the mGFR and eGFR measurements. Units: ml/min/1.73m². The central dashed line represents bias, and the dashed lines above and below represent the 95% limits of agreement.

Potential explanations for the high bias in the adjusted CKD-EPI and MDRD equations (Table 6) were also investigated in Black and White participants. Analyses were conducted on 1614 out of 1622 White participants, due to 8 participants having missing BMI data. Age was strongly associated with bias in both CKD-EPI and MDRD equations for White participants, but only with the CKD-EPI equation for Black participants. For the CKD-EPI equation, younger participants were associated with higher bias (correlation coefficient -0.1, 95% CI: -0.2 to -0.0 for White and -0.3 (-0.5 to -0.1) for Black), and for the MDRD equation in White participants, older participants were associated with higher bias (correlation coefficient 0.1, 95% CI: 0.0 to 0.2). Higher bias in the MDRD equation was also associated with male sex for White participants (correlation coefficient 3.0, 95% CI: 0.8 to 5.1). There were no associations with bias and BMI or referral source; however, sample size was very small in some categories of referral source, particularly for Black participants.

After exclusions, 2237 patients of self-reported African or Afro-Caribbean ancestry were known to nephrology services at the time of this study. Characteristics are reported in Table 7. Using adjusted eGFR-CKD-EPI 503/2081 (24.2%) patients were incorrectly classified according to eGFR criteria for CKD Stages but the proportion of patients with incorrect classification tended to decrease with disease severity; 279/2081 (12.5%) had unadjusted eGFR <20 ml/min/1.73m² compared with 205/2081 (9.2%) when adjusted eGFR was used, thus according to local policy 74/279 (26.5%) of patients eligible for RRT planning (which is recommended when eGFR <20 ml/min/1.73m²) may have had their care delayed.

**Table 4. Regression analysis for the association between ethnicity and mGFR to identify an ethnicity correction factor for our study sample, adjusted for age, sex and log(SCr).**

|  | Odds ratio (95% CI) | P-value |
| --- | --- | --- |
| **mGFR** | **1.018 (1.009 to 1.027)** | **<0.001** |
| Age | 0.966 (0.955 to 0.977) | <0.001 |
| Male (vs. female) | 0.564 (0.418 to 0.760) | <0.001 |
| Log(SCr) | 7.781 (4.529 to 13.595) | <0.001 |

**Table 5. Linear regression analysis for the association between log(SCr) and ethnicity, adjusted for mGFR, age and sex.**

| | Regression coefficient (95% CI) | Exponentiated regression coefficient (95% CI) | P-Value |
|---|---|---|---|
| **Black (vs. White)** | **0.129 (0.098 to 0.160)** | **1.138 (1.103 to 1.173)** | **<0.001** |
| mGFR | -0.011 (-0.012 to -0.010) | 0.989 (0.989 to 0.990) | <0.001 |
| Age | -0.006 (-0.007 to -0.005) | 0.994 (0.993 to 0.995) | <0.001 |
| Male (vs. female) | 0.231 (0.210 to 0.253) | 1.260 (1.234 to 1.287) | <0.001 |

## Discussion

This is one of the largest studies exploring the accuracy and impact of eGFR equations in people of African and Afro-Caribbean ancestry outside of Africa or the USA. We found that both adjusted eGFR-MDRD and eGFR-CKD-EPI equations lead to an overestimation of GFR compared to mGFR, particularly in people of black ethnicity with GFR $\geq$60 ml/min/1.73m$^2$. Removal of ethnicity adjustment for both eGFR-MDRD and eGFR-CKD-EPI significantly reduced bias and improved 30% accuracy of equations. LM Revised equation had the smallest bias for both Black and White participants, including in Black participants with mGFR<60ml/min/1.73m$^2$ in which all eGFR equation accuracy tended to be reduced. The best adjustment factor for ethnicity in our cohort was 1.018 i.e. only an increment of approximately 2% is needed to improve eGFR-CKD-EPI accuracy, which is unlikely to be of clinical importance.

Exploration of the clinical impact of use of eGFR-CKD-EPI demonstrated that approximately one in four patients had a more advanced CKD Stage when categorised by unadjusted eGFR-CKD-EPI. Similarly, approximately one in four patients may have had delayed planning for RRT.

However, estimated GFR equation accuracy did not meet required standards in White participants. An important shortcoming of this study is the inclusion of participants who were undergoing measured GFR studies for other indications. Higher bias was seen in younger White participants with CKD-EPI. The majority of these tests were performed prior to commencing chemotherapy, thus reduced muscle mass may be more pronounced in younger

**Table 6. Linear regression analysis for the association between the bias of CKD-EPI and MDRD equations and demographic characteristics in White participants, with mutual adjustment for all other variables in the table.** N = 1614.

| | CKD-EPI | | MDRD | |
|---|---|---|---|---|
| Characteristic | Linear regression coefficient (95% confidence interval) | | Linear regression coefficient (95% confidence interval) | |
| | White (N = 1614) | Black (N = 266) | White (N = 1614) | Black (N = 266) |
| **Age** | -0.1 (-0.2 to -0.0) | -0.3 (-0.5 to -0.1) | 0.1 (0.0 to 0.2) | -0.2 (-0.5 to 0.1) |
| **Male** | 0.9 (-0.6 to 2.4) | -5.1 (-10.4 to 0.2) | 3.0 (0.8 to 5.1) | -4.9 (-11.5 to 1.8) |
| **BMI** | 0.0 (-0.1 to 0.2) | -0.3 (-0.8 to 0.2) | -0.0 (-0.2 to 0.2) | -0.0 (-0.6 to 0.6) |
| **Referral source** | | | | |
| **Haematology** | Ref | Ref | Ref | Ref |
| **Oncology** | -0.6 (-6.3 to 5.2) | 2.2 (-13.3 to 17.7) | -1.3 (-9.7 to 7.0) | -2.2 (-21.7 to 17.3) |
| **Renal** | -1.4 (-3.8 to 1.0) | -0.3 (-10.4 to 9.9) | -1.8 (-5.2 to 1.7) | -0.3 (-13.1 to 12.5) |
| **Urology** | 1.6 (-4.6 to 7.8) | -17.8 (-39.2 to 3.5) | 0.6 (-8.4 to 9.6) | -21.8 (-48.8 to 5.1) |
| **ICU** | -2.9 (-23.9 to 18.1) | No data | 14.2 (-16.2 to 44.5) | No data |
| **Medicine** | 2.2 (-0.9 to 5.3) | 5.1 (-6.5 to 16.8) | 3.8 (-0.7 to 8.2) | 2.3 (-12.4 to 17.1) |
| **Paeds** | -1.9 (-6.0 to 2.3) | -8.0 (-20.6 to 4.6) | -3.5 (-9.5 to 2.6) | -6.3 (-22.1 to 9.6) |
| **Other** | 0.5 (-3.8 to 4.8) | -25.1 (-44.3 to -5.9) | 1.1 (-5.2 to 7.3) | -31.1 (-55.3 to -6.9) |
| **Unknown** | 2.2 (-3.3 to 7.8) | 2.0 (-9.0 to 13.0) | 2.2 (-5.9 to 10.2) | -1.1 (-15.0 to 12.8) |

**Table 7. Assessment of clinical impact of chronic kidney disease staging according to Kidney Disease Improving Global Outcomes (KDIGO) criteria using adjusted and unadjusted CKD-EPI-GFR equation in a cohort of people of self-reported 'Black Ethnicity' receiving nephrology care in the United Kingdom (excluding patients receiving dialysis).**

| Cohort Characteristics | N = 2237 | | |
|---|---|---|---|
| Age, mean (SD) | 57.7 (17.2) | | |
| Male, N (%) | 1102 (49.3%) | | |
| Renal Transplant N (%) | 154 (7.0%) | | |
| eGFR categories (ml/min/1.73m$^2$) [excluding patients with transplants] N (%) N = 2081 | CKD-EPI with ethnicity adjustment | CKD-EPI without ethnicity adjustment | Patients with incorrect classification using ethnicity adjustment |
| >90 | 620 (29.7%) | 410 (19.7%) | 210 (33.9%) |
| 60–89 | 496 (23.8%) | 535 (25.7%) | 171 (34.5%) |
| 30–59 | 582 (27.9%) | 662 (32.5%) | 91 (15.6%) |
| 15–29 | 264 (12.7%) | 324 (15.6%) | 31 (11.7%) |
| <15 | 121 (5.8%) | 152 (7.3%) | 0 |
| eGFR <20mls/min/1.73m$^2$ (including patients with transplants) | 205 (9.2%) | 279 (12.5%) | 74 (26.5%) |

CKD: Chronic Kidney Disease; eGFR: estimated Glomerular Filtration Rate.

participants is possible. However, higher bias was seen in older White participants with MDRD. We excluded those with albumin < 30g/l to attempt to exclude those who were catabolic or malnourished, but acknowledge this approach is imperfect.

Other limitations include the low numbers of participants with CKD and lack of information about hydration and fasting status. In addition, self-reported ancestry was used, and we were unable to explore regional differences in eGFR accuracy. Prospective data collection in controlled settings in people with and without CKD without additional indications for measured GFR assessment (e.g. malignancy) are needed, but will be costly and additional strategies to enhance recruitment of Black participants may be needed [32].

The adjustment factor for ethnicity in our cohort (1.018) was considerably lower than other studies. The MDRD adjustment factor (1.212) was derived from 197 African Americans (8% of total cohort), whereas the eGFR-CKD-EPI adjustment factor (1.159) was developed from pooled datasets including 1737 African Americans (32% of total cohort) and 384 African Americans, South Africans and African Europeans (N = 84) (10% of total cohort) in the validation cohort [33–35]. An adjustment factor of only 1.077 (95% CI, 1.042–1.113) was needed to enhance equation accuracy in an African French study which included 302 African Europeans [11]. In keeping with our findings, numerous studies of healthy adults and patients with CKD in East, Central, West and South Africa report that use of ethnicity correction factors leads to overestimation of GFR; [11,16,36–44] bias and accuracy of estimation equations compared with mGFR are improved without ethnicity correction (Table 8).

Serum creatinine concentrations are described in people of Black ethnicity with the same mGFR as people of White ethnicity [8,9], and serum creatinine concentrations are reported to increase with higher proportions of genetically determined African ancestry [48]. However, in a UK study of postpartum women there were no differences in serum creatinine concentration with ethnicity, but measured GFR was not performed [49]. Genetic analysis to determine proportion of African ancestry to guide use of ethnicity correction factors has been proposed, but will be impractical in clinical settings [48].

It has been suggested that people of Black ethnicity have a higher muscle mass than people of White ethnicity, but this has not been formally studied [8,9,12]. The mean BMI of Black participants in our cohort (28.2 kg/m$^2$) was higher than an African French cohort (26.0 kg/m$^2$)

**Table 8. Studies of measured and estimated glomerular filtration rate in Africa and Europe.**

| Author | Country | Number of Participants | Cohort | Measured GFR | Median Bias With Ethnicity Correction | Median Bias Without Ethnicity Correction | % estimates within 30% mGFR with ethnicity correction | % estimates within 30% mGFR without ethnicity correcton |
|---|---|---|---|---|---|---|---|---|
| *MDRD* | | | | | | | | |
| Agoons [45] | Cameroon | 51 | Type 2 Diabetes Mellitus Median mGFR 69.0 mL/min/1.73m | 24-hour creatinine clearance | -13.00% | -0.30% | - | - |
| Arlet [46] | African French | 64 | Sickle cell disease patients Median mGFR 112.5 mL/min/1.73m | Iohexol | Median difference in eGFR and mGFR 49.3 [24.7–64.8] | Median difference in eGFR and mGFR 19.9 [4.9–32.9] | - | - |
| Bukabau [16] | Democratic Republic of Congo | 93 | Healthy Adults Mean mGFR 92.0 ± 17.2 mL/min/1.73m | Iohexol | Median difference in eGFR and mGFR 13.6 [8.0–19.2] | Median difference in eGFR and mGFR -4.9 [-9.6; -0.2] | 79.6% | 86.0% |
| Madala [44] | South Africa | | CKD Outpatients 70.3% mGFR <60 mL/min | 99m-Tc-DTPA | eGFR <30: 39.2%; 30–59: 5.3%; >60: 19.3% | 29.2% 17.1% 38.0% | 53.3% 62.5% 35.1% | 36.1% 65.2% 68.8% |
| Moodley [36] | South Africa | 188 | Nuclear Medicine Studies | 99m-Tc-DTPA | - | Mean bias: Female 16.4%; Male 29.1% | - | Mean bias: Female 49.9%; Male 54.3% |
| Seape [15] | South Africa | 97 | Black HIV patients | 51Cr-EDTA-GFR | 38.40% | 14.20% | 43.30% | 59.80% |
| Van Deventer [38] | South Africa | 100 | Healthy Adults | 51Cr-EDTA-GFR | 27% | 5% | - | - |
| Wyatt [47] | Kenya | 99 | HIV patients | Iohexol | 18% | -3% | 73% | 83% |
| *CKD-EPI* | | | | | | | | |
| Agoons | Cameroon | 51 | Type 2 Diabetes Mellitus Median mGFR 69.0 mL/min/1.73m | 24-hour creatinine clearance | -8.5 | 1.7 | - | - |
| Arlet | African French | 64 | Sickle cell disease patients Median mGFR 112.5 mL/min/1.73m | Iohexol | Median difference in eGFR and mGFR 30.5 [16.5–44.3] | Median difference in eGFR and mGFR [-0.7; 24.8] | - | - |
| Bukabau | Democratic Republic of Congo | 93 | Healthy Adults Mean mGFR 92.0 ± 17.2 mL/min/1.73m | Iohexol | 17.20% | 2.30% | 73.10% | 81.70% |
| Flamant [11] | African French | 302 | CKD Patients Mean mGFR 57.6 mL/min/1.73m | 51Cr-EDTA-GFR | 11.90% | - | 74.80% | - |
| Moodley | South Africa | 188 | Nuclear Medicine Studies | 99m-Tc-DTPA | Female 31.5% Male 39.4% | Femle 13.5% Male 20.2% | Femle 46.7% Male 45.7% | Femle 53.3% Male 54.3% |
| Seape | South Africa | 97 | Black HIV patients | 51Cr-EDTA-GFR | 33.70% | 15.30% | 41.20% | 62.90% |
| Wyatt | Kenya | 99 | HIV patients | Iohexol | 10% | -4% | 82% | 85% |

and African studies (e.g. healthy Africans from Ivory Coast and Democratic Republic of Congo 24 kg/m²) and more comparable to African American cohorts (MDRD study: 28.7 kg/m²; AASK trial: 30.7 kg/m2) [12] suggesting that BMI may be inadequate surrogate for muscle mass. Analysis of the Chronic Renal Insufficiency Cohort (CRIC) Study, (37% Black

participants), reported that 'ethnicity correction' bias was reduced from 20% to 3.3% when body composition variables were included their CRIC GFR estimating equation [50]. Thus, to enhance accuracy of eGFR equations, it is possible that anthropometric characteristics may also need to be considered.

Dietary protein intake and catabolism has also been proposed to contribute to GFR differences between African and African American studies. Lower dietary protein intake is described in black South Africans compared to both White South Africans and African Americans [51]. However, unlike in African Americans there is evidence that Black British people tend to eat more traditional foods with lower protein, which may account for the lower adjustment factor in our cohort. Other proposed explanations include differences in tubular creatinine secretion [52,53], but these findings are not consistent [11].

Despite the LM Revised equation being derived from a White cohort [30], this equation had the smallest bias in Black participants. Actual body mass was used as 'estimated lean body mass' in our cohort, which may have been appropriate for those with other indications for GFR measurement. The FAS equation is based on serum creatinine in healthy populations normalised for age and sex but do not include ethnicity, and high rates of bias identified in our cohort may also reflect their concurrent disease state. EKFC equation estimations were less likely to overestimate GFR than FAS, in keeping with other reports, but validation in people of Black ethnicity is limited [31]. However, unlike other equations, EKFC could potentially be used in children, although was not explored in this study. People of Black ethnicity represent 7.8% of patients requiring RRT in the UK (3.0% of overall UK Black ethnicity) [54,55]. Black patients have earlier onset ESKD (56.5 v 65.8 years) [56], and reduced pre-emptively listing for transplantation compared to White patients (odds ratio, 0.43) [54,55]. However in areas of high RRT uptake, rates of CKD diagnosis are low [57], which has also been reported in the USA, and overestimation due to eGFR equations in African Americans with $\geq$60 mls/min/1.73m$^2$ has been proposed to be contributory in keeping with our findings [37,56,58–61]. In a South African Black cohort approximately one in six patients would be not recognised as having CKD Stage 3 [36], and in a US cohort up to one in three Black patients were reclassified with a more severe CKD if unadjusted eGFR equations were used [60].

The eGFR-CKD-EPI equation is currently recommended for CKD diagnosis and staging in Kidney Disease in Global Outcomes (KDIGO) and National Institute for Health and Care Excellence (NICE) guidelines including use of adjustment for ethnicity in the UK [62]. Identification of early CKD, dependent on eGFR test accuracy, in low and middle-income countries will be critical to reduce the burden of ESKD in low resource settings [63].

KDIGO guidelines highlight potential sources of error in GFR estimation equations due to 'race/ethnicity other than US and European black and white' populations [64] but others have advised caution about elimination of adjusted eGFR equations [26,65], However, recently in the USA, there has movement away from race-based medicine. Some institutions have abolished application of adjusted eGFR due to concerns about racial categorisation being used in a non-standard way including for those of mixed ancestry [65]. In 2020, the National Kidney Foundation and American Society of Nephrology established a task force to reassess the inclusion of race eGFR equations in the United States and its final recommendations for future practice are awaited.

## Conclusion

Overestimation of measured GFR with eGFR equations using ethnicity adjustment may lead to reduced rates of CKD diagnosis and under-recognition of CKD severity in people of Black ethnicity in the UK. Our findings suggest that ethnicity correction factors for GFR estimation

in non-African Americans should no longer be used, until better approaches of assessment are available. Given the consistency with data reported by other groups studying GFR in people of African ancestry outside of the USA, we consider that these findings are generalisable to other UK and European hospitals.

## Author Contributions

**Conceptualization:** Amanda Clery, Adrien M. Peters, Henry Kibble, Royce P. Vincent, Claire C. Sharpe, Hugh Cairns, Kate Bramham.

**Data curation:** Amanda Clery, Neil Heraghty.

**Formal analysis:** Amanda Clery, Kieran Palmer, Royce P. Vincent.

**Investigation:** Rouvick M. Gama, Amanda Clery, Kieran Palmer, Henry Kibble.

**Methodology:** Rouvick M. Gama, Amanda Clery, Kieran Palmer, Henry Kibble, Royce P. Vincent.

**Project administration:** Neil Heraghty.

**Resources:** Neil Heraghty, Adrien M. Peters.

**Supervision:** Adrien M. Peters.

**Writing – original draft:** Rouvick M. Gama, Amanda Clery, Henry Kibble.

**Writing – review & editing:** Rouvick M. Gama, Kathryn Griffiths, Adrien M. Peters, Royce P. Vincent, Claire C. Sharpe, Hugh Cairns, Kate Bramham.

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
