## [Decision Letter · Decision Letter 0]

12 Mar 2021

PONE-D-20-41065

Estimated Glomerular Filtration Rate Equations in Black British people:  Inappropriate adjustment for ethnicity may lead to reduced access to care

PLOS ONE

Dear Dr. Griffiths,

Thank you for submitting your manuscript to PLOS ONE. After careful consideration, we feel that it has merit but does not fully meet PLOS ONE’s publication criteria as it currently stands. Therefore, we invite you to submit a revised version of the manuscript that addresses the points raised during the review process.

We look forward to receiving your revised manuscript.

Kind regards,

Pierre Delanaye

Academic Editor

PLOS ONE

Journal Requirements:

2.We note that you state that your study was reviewed locally and was not considered to need research ethics committee approval. We do not feel that the documentation provided constitutes a waiver from an institutional ethics committee, therefore, we request that you please remove this statement. If the institutional ethics committee specifically waived the need for ethics approval, please provide the full name of the committee, and upload a copy of the letter from the ethics committee as an "Other" file.

We also note that the authors are affiliated to King's College Hospital. In your ethics statement in the manuscript methods, please explain how the laboratory databases are anonymised, and what controls are in place to prevent researchers from gaining access to potentially identifying information in the laboratory databases.

3. To comply with PLOS ONE submission guidelines, in your Methods section, please provide additional information regarding your statistical analyses. For more information on PLOS ONE's expectations for statistical reporting, please see https://journals.plos.org/plosone/s/submission-guidelines.#loc-statistical-reporting.

4. Please include your tables as part of your main manuscript and remove the individual files. Please note that supplementary tables (should remain/ be uploaded) as separate "supporting information" files.

Additional Editor Comments:

The manuscript is highly topical as the ethnic coefficient is largely debated in the literature. The sample is modest for the comparison of eGFR and mGFR.

Unfortunately, the performance of equation in White people is very low, especially in low GFR range. This is clearly a major limitation for the interpretation of the results. Such a low P30 in Whites is killing the interest of this interesting work. The authors should explain this result (and maybe try to improve): due to selection of patients (with low muscle mass?)? problem with creatinine (not IDMS)? problem with the reference method? Problem with calculation? A matched analysis (based on mGFR, age, weight or BMI, gender, referral) between Blacks and Whites might help. If no significant change is made on this part (comparison eGFR and mGFR), I will recommend the authors to focus only on the second part of the analysis (classification of patients with or without coefficient).

Also I agree with reviewer 1 on the fact that other equations should be tested (LMR and EKFC).

The discussion is too long, although the result section is too brief.

Reviewers' comments:

Reviewer's Responses to Questions

**Comments to the Author**

1. Is the manuscript technically sound, and do the data support the conclusions?

Reviewer #1: Partly

Reviewer #2: Partly

Reviewer #3: No

Reviewer #4: Yes

Reviewer #5: Yes

2. Has the statistical analysis been performed appropriately and rigorously? 

Reviewer #1: I Don't Know

Reviewer #2: Yes

Reviewer #3: No

Reviewer #4: Yes

Reviewer #5: I Don't Know

3. Have the authors made all data underlying the findings in their manuscript fully available?

Reviewer #1: No

Reviewer #2: Yes

Reviewer #3: Yes

Reviewer #4: Yes

Reviewer #5: No

4. Is the manuscript presented in an intelligible fashion and written in standard English?

Reviewer #1: Yes

Reviewer #2: Yes

Reviewer #3: Yes

Reviewer #4: Yes

Reviewer #5: Yes

5. Review Comments to the Author

Reviewer #1: The current article investigates the need to ‘correct’ eGFR-equations for European Blacks, given that such a correction factor is available for African Americans, in two well-known equations: the MDRD Study equation and the CKD-EPI equation.

1. It’s unfortunate that the authors did not take the opportunity to evaluate more recent European equations, like the FAS-equation, the LMREV-equation and the EKFC-equation. Especially, the FAS-equation has been shown to perform better than the CKD-EPI or MDRD equation in African Blacks (see Bukabau J et al, Performance of creatinine- or cystatin C–based equations to estimate glomerular filtration rate in sub-Saharan African populations, Kidney International, 2019). Bukabau et al concluded that: “In conclusion, we showed that both MDRD and CKD-EPI equations perform better in our African population when the African American ethnic factors are omitted, especially in subjects with high GFR values. FAS SCr af has the same performance as FAS SCr. Among creatinine-based equations, FAS SCr and CKD-EPI equations performed similarly, and we suggest that FAS SCr could be slightly better in patients with CKD, but these results need to be confirmed in larger African CKD cohorts.” See also Yayo E et al (your ref 17: Measured (and estimated) glomerular filtration rate: reference values in West AfricaNephrol Dial Transplant (2018) 33: 1176–1180) who concluded: “Regarding eGFR equations, our results showed the best concordance with mGFR for the FAS creatinine equation, confirming the results in Caucasian cohorts but using the Q values adapted for Africans. Regarding the CKD-EPI equation, recognized to be useful in the normal GFR range, a better fit with mGFR percentiles is observed when the ethnic coefficient is not used, confirming prior data in Africans, European Africans and even AAs”.

2. Unlike CKD-EPI and MDRD, the FAS and EKFC equations adjust for differences in Serum Creatinine generation between children and adults and between males and females, and this allowed to develop a full age spectrum equation. No (or little) differences in GFR between children and adults, or between males and females have been shown (for children older than 2 years). This is probably due to the BSA indexation. Also, differences in GFR between ethnicities have never been shown (see Yayo et al, your ref 17). Therefore, it seems to be interesting to have a better look at possible differences between European Blacks and Whites at the creatinine level, not at the GFR-level. Do the authors see systematic differences in creatinine generation between Blacks and Whites? The whole idea of ‘correcting for differences in creatinine generation’ between populations disserves more attention.

3. How was mGFR obtained? Using the full concentration-time decay curve, or using only late samples and correcting for the absence of the early compartment? Please give more details about the measurement method.

4. 11% of 2333 is not equal to 314 Blacks. Do I miss something? Or, is this the difference between self-reported Blacks and what? How did the authors define ‘Black ethnicity’ when it was not self-reported? In figure 1, the % of Blacks is 13.5%. Please check!

5. It is surprising to see the very high bias for MDRD (14.3) and CKD-EPI (14.6) in White participants. MDRD is not really the best equation to estimate GFR in “healthy” participants, because it largely overestimates GFR, but the bias of CKD-EPI is exceptionally high. Can the authors explain this? It is known that CKD-EPI largely overestimates mGFR in young adults (18-30 years), but I cannot imagine that this might be the reason here? Moreover, in patients with low Scr (Scr/k < 1, with k = 0.90 mg/dL for males and 0.70 mg/dL for females), MDRD largely overestimates GFR, which is not really the case for CKD-EPI. So, I would expect to see a larger bias for MDRD than for CKD-EPI. I would advice the authors to check their calculations! See also the articles of Bukabau and Yayo where the bias obtained with CKD-EPI (without adjustment for ethnicity) were not so large, compared to the here reported bias.

6. Also, P30 accuracy is very low for Whites. Could that be due to the Jaffe type assay, although the authors claim that SCr is IDMS equivalent? Other studies that compared eGFR equations did not show large biases for CKD-EPI (see e.g. Pottel et al. An estimated glomerular filtration rate equation for the full age spectrum. Nephrol Dialysis Transplant (2016) 31: 798-806 and Estimating glomerular filtration rate for the full age spectrum from serum creatinine and cystatin C. Nephrol Dial Transplant (2017) 32: 497–507; and the very recent Development and Validation of a Modified Full Age Spectrum Creatinine-Based Equation to Estimate Glomerular Filtration Rate: A Cross-sectional Analysis of Pooled Data. Ann Intern Med. (2020) doi:10.7326/M20-4366. ) Can the authors explain the large bias (14.6) for CKD-EPI in Whites? I am really concerned about this.

7. The CKD-EPI equation has been developed to overcome the major flaw of the MDRD equation, namely the overestimation of mGFR when mGFR > 60 mL/min/1.73m². Therefore, it was recommended to report the estimated GFR calculated from MDRD as > 60 mL/min/1.73m² instead of reporting the actual value. Thus, the authors should be careful when reporting eGFR and attributing large bias to ethnicity, rather than to the equation itself. However this does not explain the even larger bias in CKD-EPI for Whites (as compared to MDRD). Please check your calculations!

Reviewer #2: Dear Editor,

I read with interest the paper entitled ‘’ Estimated glomerular filtration rate equations in Blacks British people: inappropriate adjustement for ethnicity may led to reduced access to care’’

Ronvick Gama et al studies 2333 participants (314 black and 2019 White) and found that eGFR equations using ethnicity correction in Black British people overestimated mGFR. Although interesting, I have some several concerns.

1. Participants were recruited from hospital database; did they have acute kidney disease or CKD?

2. Authors must clarify which statistical analysis was used to evaluate the performance of equations versus mGFR? I suggest author to perform the Bland and Altman analysis in order to illustrate performance of the different equations in figure.

3. Authors do not explain if mGFR and eGFR were performed at the same time? Additional explanations are necessary for a better understanding of the paper.

4. Also what is the operational definition of Blacks British people? How long they stay in UK? Black people living in UK are they comparable to those living in Africa? After how long time can you expected to see changes in muscle mass?

5. Are the both equations comparable, especially in healthy subjects?

Finally, I think this paper needs major revisions before being published.

Reviewer #3: The present work, performed in a British population, evaluates the impact of taking into account an ethnic factor on the performance of GFR estimation formulae (eGFR), and its consequences on the management of end-stage kidney disease, through the prism of the eligibility for kidney transplantation. The study is based on 2333 EDTA clearances (mGFR) performed over 10 years, of which 334 were obtained in black patients. This work describes a dramatic overestimation of mGFR in patients of African origin when the ethnic factor is taken into account, with a significant improvement in accuracy when the ethnic factor is not taken into account. In a second part of the work, conducted in a cohort of patients of 2237 patients of African origin, the authors evaluate that 26 % of patients had unadjusted eGFR below 20 ml/min/1.73m² and may have been delayed for RRT planning when an ethnic factor is considered. Taken together, the authors conclude that the ethnic factor should not be taken into account when estimating GFR with either MDRD or CKD-EPI in the British black population.

The need to consider an ethnic factor for GFR estimation in patients of African origin other than that which was used to establish the estimation formulas is an undoubtedly major and interesting issue. Unfortunately, this work has many methodological limitations leading the authors to conclude in a totally contradictory way with their data.

Major concerns

. The question underlying this study is the risk of inappropriate management of CKD patients whith the use of an ethnic-adjusted eGFR. While the number of subjects of African origin is significant in this study, the subpopulation of patients with a GFR less than 60ml/min/1.73m² is weak and insufficient to be able to draw any conclusion for the main goal of this work (n=56).

. A major issue of this study is that eGFR dramatically overestimated mGFR both in the white and in the black populations. This overestimation greatly exceeds that observed in all studies that compared the estimators to a reference method, including with the 51CrEDTA tracer. Very importantly, this is not only the case for black patients (when the ethnic factor is incorporated), but also in white patients. This strongly suggests a major issue on the assessment of reference values in this work. This is all the more problematic as all the conclusions of this work are exclusively based on the bias between adjusted-eGFR and mGFR in the black population.

These data appear to strongly support the need for an ethnic factor, in contrast with the conclusion of this work. Indeed, in the whole population, it turns out that although mGFR is 3,5 ml/min higher in the black population than in white patients, non adjusted-CKDEPI is 5 ml lower, which highlights the need for a correction factor. In other words, despited a higher mGFR, black patients have a higher creatininemia (lower eGFR), demonstrating an influence of ethnicity on serum creatinine level, independent of mGFR. Inon-adjusted eGFR assesses the difference in true GFR between the two populations with an error of nearly 8.5ml/min. This error is “only” 5.5ml/min with adjusted CKDEPI (+9ml/min/1.73m² versus +3.5ml/min/1.73m²). Altogether, this seems to call for an intermediate correction factor for ethnicity, but the lack of a correction factor leads to a larger error than the use of the existing factors.

The concern is even worse when considering patients whose GFR is less than 60ml / min / 1.73m². In this subpopulation, the mean biases between adjusted-eGFRs and mGFR in the black population are very close to those obtained in the white population (respectively 15 and 16 ml/min/1.73m² for the CKDEPI equation and 17 and 18 ml/min/1.73m² for the MDRD equation). Consequently, not taking into account the ethnic factor in the black population leads to very important differences in the mean biases between eGFR and mGFR between black and whites.

The conclusions of the authors therefore appear to be in string contradiction with the data, which actually strongly suggest the need to use an ethnic factor in this black population, although the correction factor to be applied seems lower than that proposed for African Americans. Interestingly and not previously described, this factor could be different depending on GFR value.

In a general way, the interest of including an ethnic factor for GFR estimation in a population can only be achieved by comparing the estimates between two populations that differ only by ethnic status, or alternatively by a regression model evaluating whether this status is a factor independently associated with serum creatinine value. In any case, these evaluations must be methodologically independent of the reference value, namely mGFR.

. The part of the study evaluating the risk of misclassification of black patients according to adjusted or non-adjusted eGFR is also a matter of concern, as unadjusted eGFR is considered as the reference method, in relation with the conclusions of the first part of the study. It is also important to note that the impact of the ethnic factor could have been obtained theoretically and independently of any data collection.

Minor concerns

. Bias is defined as mGFR minus mGFR in methods but results discussed in text and implemented in tables seem to indicate otherwise

. Were there repeated GFR measurements in the same patient, which is not uncommon for this type of assessment, especially in transplant patients? The flow chart does not seem to indicate any data exclusion related to repeated measures. In other words, are there 2333 different patients or 2333 different GFR measurements? The number of patients should be indicated if several measurements arise from the same patients, or ideally, only one visit per patient should be kept in the analysis.

. Given the very unusual difference between mGFR and eGFR, it would have been interesting to have some methodological details on the measurement of GFR (Plasma clearance or unirany clearance? Single point method? Equation used for the correction of the plasma clearance...)

Reviewer #4: The authors investigate the accuracy of using the Black race coefficient among Black Europeans versus White Europeans by comparing measured GFR with MDRD and CKD-EPI eGFR estimates. They demonstrate the use of the Black race coefficient significantly overestimates kidney function among Black Europeans compared to White Europeans. The authors suggest that the Black race coefficient should not be used in Europe. This study nicely complements growing evidence globally that the Black race coefficient results in overestimation of kidney function. A few suggestions and clarifications may help strengthen this manuscript:

Results:

Paragraph 5, first sentence: "After exclusion, 37 patients..." It is not clear which type of patients are excluded here. Please clarify.

Paragraph 5, last sentence: It is not clear how delayed RRT planning is determined here. Was this assessed over time after removal of the Black race factor? More details are needed here.

Discussion:

In general, this Discussion is too long and not focused. There is too much background about eGFR studies - it almost read as a review. It would be helpful to explain the findings of overestimation in GFR between Black and White patients including inaccuracies surrounding GFR estimation and measurement more generally. As it reads, the Discussion focuses on why Black individuals may have different genetic characteristics that could explain variability in eGFR (based on muscle mass) compared to other races, however the evidence that supports this is vert poor. There is discussion about socioeconomic differences between Black individuals and other races however this needs to be better organized.

Paragraph 6, 2nd sentence: Udler et al study that is referenced has NOT confirmed association of higher muscle mass based on African ancestry. To my knowledge, no study has done this. Please explain this sentence.

Paragraph 7, 3rd sentence: "Whilst it might be assumed that African Americans and Europeans have similar diet..." Why would it be assumed that African Americans and Europeans have similar diet? More details are needed here.

Paragraph 8, 2nd sentence: "Thus, to enhance accuracy of eGFR equations for people of African and Afro-Caribbean ancestry.." Why would accuracy only be enhanced for people of African and Afro-Caribbean ancestry? What about other races? Please expand here.

Reviewer #5: Thank you for the opportunity to review this manuscript, which is highly topical, and this month the American Society of Nephrology announced they were abandoning racial adjustment for eGFR. Authors should reference this and the informing literature published in CJASN this year (https://cjasn.asnjournals.org/content/early/2021/03/04/CJN.01780221)

This manuscript potentially takes a more precise approach than some of the papers published recently exploring if the adjustment in the two equations should be dropped, not because this improves their accuracy, but rather that their removal would initiate pre-dialysis planning sooner in this group who for a range of reasons do not have equitable access to the best possible healthcare. Indeed removal for race has been cited as inappropriate (https://cjasn.asnjournals.org/content/15/8/1203)

This manuscript does require some additional work to ensure the community gets the most possible from it:

Introduction - please mention creatinine (which one would consider the reason race is being adjusted for although this is explored in the discussion) earlier.

Please acknowledge some of the policy decisions which are being suggested around race adjustment

Methods -

I need to linger on the mean difference and associated precision: as written I was not able to establish if the value used here could only be positive, or could be positive or negative. i.e was this the root mean square error or just the error (presumably eGFR - mGFR as black patients had their kidney function overestimated)? This difference is fairly important, as it give some vital context as to why the SD of the error/bias is so large (same size as the error itself in many circumstances). The SD of the bias is an attempt to give the reader an appreciation of the distribution of the error, so we are saying that 68% of the error data for CKDepi overall lies between -2 (20-21.7) and 42 (20+21.7) if we were using values with a sign I believe? The limits of agreement are effectively the range across which 95% of the data lies, but one cannot put full trust in this without knowing what the original bias value (on which all these numbers rely) was derived on.

Some minor comments on the methods: one would normally have a section specifically on the statistical methods. Data processing would normally come before this.

Results -

Could we please see:

1. Some graphs comparing the data: Histograms of the mGFR and eGFR for the two different race groups for instance? Histograms of the bias ( I believe preferably with pos/neg values rather than RMSE). A lowess smoothing plot of the bias for both ethnic groups against eGFR which might cope a bit better with the small numbers in some CKD categories.

2. Again, understanding reliably the direction of the change is important: for instance when reporting the proportions who change CKD stage, this should probably be divided into % higher and % lower. 1.1ml bias (first line page 11 on my version - why aren't your pages numbered?) is very low - again knowing the RMSE around this would help know if you've improved the error on one size but worsened it on the other.

Discussion -

Clearly a major source of "bias" in the existing equations are the differences in the cohorts which they have been derived and then applied. This is acknowledged as a limitation but not really explored. For instance, what were the BMIs of the seminal papers and how do they compare to your cohort? These are well described.

The discussion is rather long (3.5 pages without line spacing) and probably has elements which could be sacrificed (e.g. you talk about Cockcroft-Gault). A lot of what is mentioned is context and not how your research aligns with existing findings (example: 2nd paragraph page 13 leading with "in the UK, the prevalence" - this could again be sacrificed in the discussion, and mentioned in the introduction). Can I suggest more formally structuring this around: a) Summary of findings b) How findings compare with existing research and any mechanisms you wish to mention c) Strengths and weaknesses d) policy and practice implications e) Recommendations for future research f) Conclusion.

6. PLOS authors have the option to publish the peer review history of their article (what does this mean?). If published, this will include your full peer review and any attached files.

Reviewer #1: No

Reviewer #2: No

Reviewer #3: No

Reviewer #4: No

Reviewer #5: No

---

## [Author Response · Author response to Decision Letter 0]

21 Jun 2021

2.We note that you state that your study was reviewed locally and was not considered to need research ethics committee approval. We do not feel that the documentation provided constitutes a waiver from an institutional ethics committee, therefore, we request that you please remove this statement. If the institutional ethics committee specifically waived the need for ethics approval, please provide the full name of the committee, and upload a copy of the letter from the ethics committee as an "Other" file.

Thank you for this recommendation. We have removed the statement from the submission. 

We also note that the authors are affiliated to King's College Hospital. In your ethics statement in the manuscript methods, please explain how the laboratory databases are anonymised, and what controls are in place to prevent researchers from gaining access to potentially identifying information in the laboratory databases.

Data were extracted by clinical staff, and given a unique study ID, all potential identifiers (name, date of birth and hospital number were removed) prior to analysis by the research team. All Non-clinical staff do not have access to any laboratory records.

3. To comply with PLOS ONE submission guidelines, in your Methods section, please provide additional information regarding your statistical analyses. For more information on PLOS ONE's expectations for statistical reporting, please see https://journals.plos.org/plosone/s/submission-guidelines.#loc-statistical-reporting.

Thank you for this recommendation. We have provided the following additional information regarding our statistical analysis.

4. Please include your tables as part of your main manuscript and remove the individual files. Please note that supplementary tables (should remain/ be uploaded) as separate "supporting information" files.

Tables have now been added as part of the main manuscript.

Editors Comments

The manuscript is highly topical as the ethnic coefficient is largely debated in the literature. The sample is modest for the comparison of eGFR and mGFR.

Unfortunately, the performance of equation in White people is very low, especially in low GFR range. This is clearly a major limitation for the interpretation of the results. Such a low P30 in Whites is killing the interest of this interesting work. The authors should explain this result (and maybe try to improve): due to selection of patients (with low muscle mass?)? problem with creatinine (not IDMS)? problem with the reference method? Problem with calculation? A matched analysis (based on mGFR, age, weight or BMI, gender, referral) between Blacks and Whites might help. If no significant change is made on this part (comparison eGFR and mGFR), I will recommend the authors to focus only on the second part of the analysis (classification of patients with or without coefficient).

Thank you for this comment – we have now undertaken a posthoc analayis to explore the bias in White participants, and identified that age was strongly associated with bias for both CKD-EPI and MDRD equations (younger patients higher bias with CKD-EPI, and older patients higher bias with MDRD). These data are now reported on Page XX

Also I agree with reviewer 1 on the fact that other equations should be tested (LMR and EKFC).

Thank you for this helpful suggestion. We have now tested LMR, FAS and EKFC equations. In White patients eGFR FAS equation had the greatest bias of 15.2ml/min/1.73m2. The LM Revised equation had the smallest bias, 5.8ml/min/1.73m2. In Black patients, the LM Revised had the smallest bias (-1.1ml/min/1.73m2). 

Page 9 Line 5

The discussion is too long, although the result section is too brief.

Thank you, we have now enhanced the results section and reduced the discussion section as recommended.

Reviewers' comments:

Reviewer #1: The current article investigates the need to ‘correct’ eGFR-equations for European Blacks, given that such a correction factor is available for African Americans, in two well-known equations: the MDRD Study equation and the CKD-EPI equation.

1. It’s unfortunate that the authors did not take the opportunity to evaluate more recent European equations, like the FAS-equation, the LMREV-equation and the EKFC-equation. Especially, the FAS-equation has been shown to perform better than the CKD-EPI or MDRD equation in African Blacks (see Bukabau J et al, Performance of creatinine- or cystatin C–based equations to estimate glomerular filtration rate in sub-Saharan African populations, Kidney International, 2019). Bukabau et al concluded that: “In conclusion, we showed that both MDRD and CKD-EPI equations perform better in our African population when the African American ethnic factors are omitted, especially in subjects with high GFR values. FAS SCr af has the same performance as FAS SCr. Among creatinine-based equations, FAS SCr and CKD-EPI equations performed similarly, and we suggest that FAS SCr could be slightly better in patients with CKD, but these results need to be confirmed in larger African CKD cohorts.” See also Yayo E et al (your ref 17: Measured (and estimated) glomerular filtration rate: reference values in West AfricaNephrol Dial Transplant (2018) 33: 1176–1180) who concluded: “Regarding eGFR equations, our results showed the best concordance with mGFR for the FAS creatinine equation, confirming the results in Caucasian cohorts but using the Q values adapted for Africans. Regarding the CKD-EPI equation, recognized to be useful in the normal GFR range, a better fit with mGFR percentiles is observed when the ethnic coefficient is not used, confirming prior data in Africans, European Africans and even AAs”.

Thank you for this suggestion. We have now revised the analysis to included the FAS-equation, the LMR-equation and the EKFC-equations and found that overall LMR had least bias in both Black and White patients.

Please see Page 9 Line 5.

2. Unlike CKD-EPI and MDRD, the FAS and EKFC equations adjust for differences in Serum Creatinine generation between children and adults and between males and females, and this allowed to develop a full age spectrum equation. No (or little) differences in GFR between children and adults, or between males and females have been shown (for children older than 2 years). This is probably due to the BSA indexation. Also, differences in GFR between ethnicities have never been shown (see Yayo et al, your ref 17). Therefore, it seems to be interesting to have a better look at possible differences between European Blacks and Whites at the creatinine level, not at the GFR-level. Do the authors see systematic differences in creatinine generation between Blacks and Whites? The whole idea of ‘correcting for differences in creatinine generation’ between populations disserves more attention.

We have now compared creatinine level in Black and White ethnicities, matched for GFR and found higher creatinine concentrations. We agree this warrants futher focus and have introduced this into the discussion.

3. How was mGFR obtained? Using the full concentration-time decay curve, or using only late samples and correcting for the absence of the early compartment? Please give more details about the measurement method.

mGFR was measured by Cr51-EDTA, administered iv (10 MBq), was used to measure mGFR. Plasma clearance of the tracer was calculated from accurately-timed plasma samples obtained at about 120, 180 and 240 min, and corrected for the assumption of a single compartment using the formula of Brochner-Mortensen

Page 4 Line 10

4. 11% of 2333 is not equal to 314 Blacks. Do I miss something? Or, is this the difference between self-reported Blacks and what? How did the authors define ‘Black ethnicity’ when it was not self-reported? In figure 1, the % of Blacks is 13.5%. Please check!

We have reviewed our calculations and after removal of serial testing, confirm that 266 out of 1888 (14.1%) of the participants were self-reported black ethnicity.

Only self-reported black ethnicity was used. If no ethnicity was reported participants were excluded.

5. It is surprising to see the very high bias for MDRD (14.3) and CKD-EPI (14.6) in White participants. MDRD is not really the best equation to estimate GFR in “healthy” participants, because it largely overestimates GFR, but the bias of CKD-EPI is exceptionally high. Can the authors explain this? It is known that CKD-EPI largely overestimates mGFR in young adults (18-30 years), but I cannot imagine that this might be the reason here? Moreover, in patients with low Scr (Scr/k < 1, with k = 0.90 mg/dL for males and 0.70 mg/dL for females), MDRD largely overestimates GFR, which is not really the case for CKD-EPI. So, I would expect to see a larger bias for MDRD than for CKD-EPI. I would advice the authors to check their calculations! See also the articles of Bukabau and Yayo where the bias obtained with CKD-EPI (without adjustment for ethnicity) were not so large, compared to the here reported bias.

We have reviewed our calculations and confirm that age was strongly associated with bias in both CKD-EPI and MDRD equations for White patients, but only with the CKD-EPI equation for Black patients. For the CKD-EPI equation, younger patients were associated with higher bias (correlation coefficient -0.1, 95% CI: -0.2 to -0.0 for White and -0.3 (-0.5 to -0.1) for Black), and for the MDRD equation in White patients, older patients were associated with higher bias (correlation coefficient 0.1, 95% CI: 0.0 to 0.2). Higher bias in the MDRD equation was also associated with male sex for White patients (correlation coefficient 3.0, 95% CI: 0.8 to 5.1). There were no associations with bias and BMI or referral source; however sample size was very small in some categories of referral source, particularly for Black patients. Page 11 Line 6-16

6. Also, P30 accuracy is very low for Whites. Could that be due to the Jaffe type assay, although the authors claim that SCr is IDMS equivalent? Other studies that compared eGFR equations did not show large biases for CKD-EPI (see e.g. Pottel et al. An estimated glomerular filtration rate equation for the full age spectrum. Nephrol Dialysis Transplant (2016) 31: 798-806 and Estimating glomerular filtration rate for the full age spectrum from serum creatinine and cystatin C. Nephrol Dial Transplant (2017) 32: 497–507; and the very recent Development and Validation of a Modified Full Age Spectrum Creatinine-Based Equation to Estimate Glomerular Filtration Rate: A Cross-sectional Analysis of Pooled Data. Ann Intern Med. (2020) doi:10.7326/M20-4366. ) Can the authors explain the large bias (14.6) for CKD-EPI in Whites? I am really concerned about this.

We have repeated our analysis and confirm that P30 is low, especially in those with CKD, and was explained by a higher bias in younger white patients. This may reflect the comorbidities of patients undergoing mGFR testing, and relative reduction in muscle mass may be more exaggerated in younger patients (e.g. those commencing chemotherapy). The FAS equation appeared to perform better than CKD EPI in those with CKD.

7. The CKD-EPI equation has been developed to overcome the major flaw of the MDRD equation, namely the overestimation of mGFR when mGFR > 60 mL/min/1.73m². Therefore, it was recommended to report the estimated GFR calculated from MDRD as > 60 mL/min/1.73m² instead of reporting the actual value. Thus, the authors should be careful when reporting eGFR and attributing large bias to ethnicity, rather than to the equation itself. However this does not explain the even larger bias in CKD-EPI for Whites (as compared to MDRD). Please check your calculations!

Thank you for these comments – please see response to comment 6.

Reviewer #2: Dear Editor,

I read with interest the paper entitled ‘’ Estimated glomerular filtration rate equations in Blacks British people: inappropriate adjustement for ethnicity may led to reduced access to care’’

Ronvick Gama et al studies 2333 participants (314 black and 2019 White) and found that eGFR equations using ethnicity correction in Black British people overestimated mGFR. Although interesting, I have some several concerns.

1. Participants were recruited from hospital database; did they have acute kidney disease or CKD?

Participants were retrospectively selected from hospital databases. They did not have acute kidney injury but some did have CKD Stages 3-5. We were unable to identify if individuals had CKD Stages 1 or 2 due to incomplete proteinuria/haematuria imaging assessments. This detail has now been clarified in the methods.

2. Authors must clarify which statistical analysis was used to evaluate the performance of equations versus mGFR? I suggest author to perform the Bland and Altman analysis in order to illustrate performance of the different equations in figure.

Thank you for this suggestion. We have now included further details about the statistical analysis in the methods and included a Bland Altman analysis (Figure 4) (Page 10 Line 11).

3. Authors do not explain if mGFR and eGFR were performed at the same time? Additional explanations are necessary for a better understanding of the paper.

Serum creatinine (IDMS traceable assay) taken within one week of 51Cr-EDTA study were used to calculate eGFR. Methods Page 3 Line 37.

4. Also what is the operational definition of Blacks British people? How long they stay in UK? Black people living in UK are they comparable to those living in Africa? After how long time can you expected to see changes in muscle mass?

We have changed the title to ‘in People of Self-Reported Black Ethnicity’ to avoid confusion about place of birth and the definition of ‘Black British’. We were unable to assess how long individuals have been living in the UK. Only self-reported ethnicity were available and no additional details about country of birth or ancestry were available, therefore we are unable to address influences of genetics and environment on muscle mass or GFR. 

5. Are the both equations comparable, especially in healthy subjects?

Unfortunately we are not able to identify ‘healthy’ subjects in our cohort. However in those with GFR >60mls/min LMR and EKFC were the best performing equations in both White and Black ethnicities, and CKD-EPI performed better than MDRD. (Table 3).

Reviewer #3: The present work, performed in a British population, evaluates the impact of taking into account an ethnic factor on the performance of GFR estimation formulae (eGFR), and its consequences on the management of end-stage kidney disease, through the prism of the eligibility for kidney transplantation. The study is based on 2333 EDTA clearances (mGFR) performed over 10 years, of which 334 were obtained in black patients. This work describes a dramatic overestimation of mGFR in patients of African origin when the ethnic factor is taken into account, with a significant improvement in accuracy when the ethnic factor is not taken into account. In a second part of the work, conducted in a cohort of patients of 2237 patients of African origin, the authors evaluate that 26 % of patients had unadjusted eGFR below 20 ml/min/1.73m² and may have been delayed for RRT planning when an ethnic factor is considered. Taken together, the authors conclude that the ethnic factor should not be taken into account when estimating GFR with either MDRD or CKD-EPI in the British black population.

The need to consider an ethnic factor for GFR estimation in patients of African origin other than that which was used to establish the estimation formulas is an undoubtedly major and interesting issue. Unfortunately, this work has many methodological limitations leading the authors to conclude in a totally contradictory way with their data.

Major concerns

1. The question underlying this study is the risk of inappropriate management of CKD patients with the use of an ethnic-adjusted eGFR. While the number of subjects of African origin is significant in this study, the subpopulation of patients with a GFR less than 60ml/min/1.73m² is weak and insufficient to be able to draw any conclusion for the main goal of this work (n=56).

We recognise the limitation of the small number of patients with CKD in the cohort, which is limited by its retrospective design. However, we feel that exploration of accuracy of eGFR equations remains important in those with eGFR over 60mls/min/1.73m2, as this remains the standard method to diagnose CKD Stage 3 in the UK.

2. A major issue of this study is that eGFR dramatically overestimated mGFR both in the white and in the black populations. This overestimation greatly exceeds that observed in all studies that compared the estimators to a reference method, including with the 51CrEDTA tracer. Very importantly, this is not only the case for black patients (when the ethnic factor is incorporated), but also in white patients. This strongly suggests a major issue on the assessment of reference values in this work. This is all the more problematic as all the conclusions of this work are exclusively based on the bias between adjusted-eGFR and mGFR in the black population.

We recognise that the bias was high for both black and white participants, and have revised our conclusions accordingly.

3. These data appear to strongly support the need for an ethnic factor, in contrast with the conclusion of this work. Indeed, in the whole population, it turns out that although mGFR is 3,5 ml/min higher in the black population than in white patients, non adjusted-CKDEPI is 5 ml lower, which highlights the need for a correction factor. In other words, despited a higher mGFR, black patients have a higher creatininemia (lower eGFR), demonstrating an influence of ethnicity on serum creatinine level, independent of mGFR. Inon-adjusted eGFR assesses the difference in true GFR between the two populations with an error of nearly 8.5ml/min. This error is “only” 5.5ml/min with adjusted CKDEPI (+9ml/min/1.73m² versus +3.5ml/min/1.73m²). Altogether, this seems to call for an intermediate correction factor for ethnicity, but the lack of a correction factor leads to a larger error than the use of the existing factors.

Thank you for this helpful suggestion. We have now undertaken further analysis including identification of a higher ‘creatininemia’ and calculation of a ethnicity correction factor which was only 1.018.

4. The concern is even worse when considering patients whose GFR is less than 60ml / min / 1.73m². In this subpopulation, the mean biases between adjusted-eGFRs and mGFR in the black population are very close to those obtained in the white population (respectively 15 and 16 ml/min/1.73m² for the CKDEPI equation and 17 and 18 ml/min/1.73m² for the MDRD equation). Consequently, not taking into account the ethnic factor in the black population leads to very important differences in the mean biases between eGFR and mGFR between black and whites.

We agree with the reviewer’s comments that performance in considerably worse in both White and Black patients with lower GFR, and have modified the discussion accordingly to encourage recognition of the inaccuracies of equations in advanced CKD.

5. The conclusions of the authors therefore appear to be in string contradiction with the data, which actually strongly suggest the need to use an ethnic factor in this black population, although the correction factor to be applied seems lower than that proposed for African Americans. Interestingly and not previously described, this factor could be different depending on GFR value.

In a general way, the interest of including an ethnic factor for GFR estimation in a population can only be achieved by comparing the estimates between two populations that differ only by ethnic status, or alternatively by a regression model evaluating whether this status is a factor independently associated with serum creatinine value. In any case, these evaluations must be methodologically independent of the reference value, namely mGFR.

Thank you for this suggestion. We have now undertaken further analysis and confirmed that ethnicity is independently associated with serum creatinine, and reported an ethnicity factor for this cohort. (Page 10 Line 19)

6. The part of the study evaluating the risk of misclassification of black patients according to adjusted or non-adjusted eGFR is also a matter of concern, as unadjusted eGFR is considered as the reference method, in relation with the conclusions of the first part of the study. It is also important to note that the impact of the ethnic factor could have been obtained theoretically and independently of any data collection.

Need to consider re-assessing the clinical impact with new correction factor.

Minor concerns

7. Bias is defined as mGFR minus mGFR in methods but results discussed in text and implemented in tables seem to indicate otherwise

NEED TO CHECK

8. Were there repeated GFR measurements in the same patient, which is not uncommon for this type of assessment, especially in transplant patients? The flow chart does not seem to indicate any data exclusion related to repeated measures. In other words, are there 2333 different patients or 2333 different GFR measurements? The number of patients should be indicated if several measurements arise from the same patients, or ideally, only one visit per patient should be kept in the analysis.

We have now repeated the analysis and only the first mGFR only was used and repeated measurements excluded. This detail is clarified in Methods: Page 3 Line 38.

9. Given the very unusual difference between mGFR and eGFR, it would have been interesting to have some methodological details on the measurement of GFR (Plasma clearance or unirany clearance? Single point method? Equation used for the correction of the plasma clearance...)

mGFR was measured by Cr51-EDTA, administered iv (10 MBq), was used to measure mGFR. Plasma clearance of the tracer was calculated from accurately-timed plasma samples obtained at about 120, 180 and 240 min, and corrected for the assumption of a single compartment using the formula of Brochner-Mortensen. (Page 4 Line 7)

Reviewer #4: The authors investigate the accuracy of using the Black race coefficient among Black Europeans versus White Europeans by comparing measured GFR with MDRD and CKD-EPI eGFR estimates. They demonstrate the use of the Black race coefficient significantly overestimates kidney function among Black Europeans compared to White Europeans. The authors suggest that the Black race coefficient should not be used in Europe. This study nicely complements growing evidence globally that the Black race coefficient results in overestimation of kidney function. A few suggestions and clarifications may help strengthen this manuscript:

Results:

Paragraph 5, first sentence: "After exclusion, 37 patients..." It is not clear which type of patients are excluded here. Please clarify.

Thank you for this suggestion we have now added the following:

After exclusion of participants with predefined confounders which might influence serum creatinine concentration, a total of…… (Page 5 line 34)

Paragraph 5, last sentence: It is not clear how delayed RRT planning is determined here. Was this assessed over time after removal of the Black race factor? More details are needed here.

Thank you for this recommendation we have now clarified as follows:

74/279 (26.5%) had unadjusted eGFR <20 ml/min/1.73m2 and according to local policy may have been delayed for RRT planning (which is recommended when eGFR <20 ml/min/1.73m2)

Discussion:

In general, this Discussion is too long and not focused. There is too much background about eGFR studies - it almost read as a review. It would be helpful to explain the findings of overestimation in GFR between Black and White patients including inaccuracies surrounding GFR estimation and measurement more generally. As it reads, the Discussion focuses on why Black individuals may have different genetic characteristics that could explain variability in eGFR (based on muscle mass) compared to other races, however the evidence that supports this is vert poor. There is discussion about socioeconomic differences between Black individuals and other races however this needs to be better organized.

Thank you for this suggestion. We have now shortened and refocussed the discussion, with removal of the background of eGFR equation derivation.

Paragraph 6, 2nd sentence: Udler et al study that is referenced has NOT confirmed association of higher muscle mass based on African ancestry. To my knowledge, no study has done this. Please explain this sentence.

Thank you for requesting this clarification. This inaccuracy has now been amended to clarify that ‘serum creatinine concentrations are reported to increase with higher proportions of genetically determined African ancestry.’

Paragraph 7, 3rd sentence: "Whilst it might be assumed that African Americans and Europeans have similar diet..." Why would it be assumed that African Americans and Europeans have similar diet? More details are needed here.

We have clarified this statement and supported with references outlining difference in diet between African Americans and Europeans.

Whilst it might be assumed that both African Americans and Europeans adopt a local diet; however, unlike in African Americans there is evidence that Black British people tend to eat more traditional foods. Page 14 Line 12.

Paragraph 8, 2nd sentence: "Thus, to enhance accuracy of eGFR equations for people of African and Afro-Caribbean ancestry.." Why would accuracy only be enhanced for people of African and Afro-Caribbean ancestry? What about other races? Please expand here.

Thank you for this suggestion. We agree that it is not just for people of African and Afro Caribbean ancestry that could benefit from additional anthropometric assessment to enhance eGFR accuracy and have removed this suggestion.

Reviewer #5: Thank you for the opportunity to review this manuscript, which is highly topical, and this month the American Society of Nephrology announced they were abandoning racial adjustment for eGFR. Authors should reference this and the informing literature published in CJASN this year (https://cjasn.asnjournals.org/content/early/2021/03/04/CJN.01780221)

Thank you for this suggestion. We have now referenced these publications which were not available at the original time of submission. 

This manuscript potentially takes a more precise approach than some of the papers published recently exploring if the adjustment in the two equations should be dropped, not because this improves their accuracy, but rather that their removal would initiate pre-dialysis planning sooner in this group who for a range of reasons do not have equitable access to the best possible healthcare. Indeed removal for race has been cited as inappropriate (https://cjasn.asnjournals.org/content/15/8/1203)

This manuscript does require some additional work to ensure the community gets the most possible from it:

Introduction - please mention creatinine (which one would consider the reason race is being adjusted for although this is explored in the discussion) earlier.

The issues related to creatinine excretion and race are now highlighted in the introduction as follows.

ADD.

Please acknowledge some of the policy decisions which are being suggested around race adjustment

Thank you for this suggestion. We have now included these policy decisions in both introduction and discussion as follows:

ADD.

Methods -

I need to linger on the mean difference and associated precision: as written I was not able to establish if the value used here could only be positive, or could be positive or negative. i.e was this the root mean square error or just the error (presumably eGFR - mGFR as black patients had their kidney function overestimated)? This difference is fairly important, as it give some vital context as to why the SD of the error/bias is so large (same size as the error itself in many circumstances). The SD of the bias is an attempt to give the reader an appreciation of the distribution of the error, so we are saying that 68% of the error data for CKDepi overall lies between -2 (20-21.7) and 42 (20+21.7) if we were using values with a sign I believe? The limits of agreement are effectively the range across which 95% of the data lies, but one cannot put full trust in this without knowing what the original bias value (on which all these numbers rely) was derived on.

We confirm that the bias could be positive or negative and have clarified further in the methods and reported in histograms and Bland Altman plots (Figure 3 and 4).

Some minor comments on the methods: one would normally have a section specifically on the statistical methods. Data processing would normally come before this.

The methods session has been expanded to give more details regarding statistical methods (Page 5 Line 3).

Results -

Could we please see:

1. Some graphs comparing the data: Histograms of the mGFR and eGFR for the two different race groups for instance? Histograms of the bias ( I believe preferably with pos/neg values rather than RMSE). A lowess smoothing plot of the bias for both ethnic groups against eGFR which might cope a bit better with the small numbers in some CKD categories.

Thank you for this helpful recommendation, which we have now added as Figures 2 and 3. We felt that the Bland Altmann plots demonstrated bias more clearly but would be happyto add a Lowess smoothing plot if the editor requires.

2. Again, understanding reliably the direction of the change is important: for instance when reporting the proportions who change CKD stage, this should probably be divided into % higher and % lower. 1.1ml bias (first line page 11 on my version - why aren't your pages numbered?) is very low - again knowing the RMSE around this would help know if you've improved the error on one size but worsened it on the other.

The proportion of change of CKD Stage refers only those who have a lower CKD stage than previously considered with use of the adjustment factor. We apologise for not having page numbers and have now added.

Discussion -

Clearly a major source of "bias" in the existing equations are the differences in the cohorts which they have been derived and then applied. This is acknowledged as a limitation but not really explored. For instance, what were the BMIs of the seminal papers and how do they compare to your cohort? These are well described.

Need to add 

The discussion is rather long (3.5 pages without line spacing) and probably has elements which could be sacrificed (e.g. you talk about Cockcroft-Gault). A lot of what is mentioned is context and not how your research aligns with existing findings (example: 2nd paragraph page 13 leading with "in the UK, the prevalence" - this could again be sacrificed in the discussion, and mentioned in the introduction). Can I suggest more formally structuring this around: a) Summary of findings b) How findings compare with existing research and any mechanisms you wish to mention c) Strengths and weaknesses d) policy and practice implications e) Recommendations for future research f) Conclusion.

Thank you for this recommendation. The discussion has now been revised and shortened, and we hope focusses more succinctly on the findings, comparison with other research etc. as recommended.

---

## [Decision Letter · Decision Letter 1]

16 Jul 2021

PONE-D-20-41065R1

Estimated Glomerular Filtration Rate Equations in people of self-reported Black ethnicity in the United Kingdom:  Inappropriate adjustment for ethnicity may lead to reduced access to care

PLOS ONE

Dear Dr. Bramham,

Thank you for submitting your manuscript to PLOS ONE. After careful consideration, we feel that it has merit but does not fully meet PLOS ONE’s publication criteria as it currently stands. Therefore, we invite you to submit a revised version of the manuscript that addresses the points raised during the review process.

We look forward to receiving your revised manuscript.

Kind regards,

Pierre Delanaye

Academic Editor

PLOS ONE

Journal Requirements:

Additional Editor Comments (if provided):

The article has been largely improved. The authors should be congratulated for that.

I have three minor comments

1) The authors wrote: "This is the largest study exploring the accuracy and impact of eGFR equations in people of African and Afro-Caribbean ancestry outside of Africa or the USA". This is incorrect. Flamant et al (ref 11) studied more European black subjects (n=302) than in the current analysis (n=266). Please tone down.

2) I agree the authors keep their coefficient, even if I also agree with Reviewer that correction should be at the creatinine level, not the GFR level. However, I question the clinical relevance of the correction 1.018. Is it really significant from a clinical perspective? To be discussed.

3) It must be reminded that the EKFC equation has the advantage to estimate GFR with the same equation in adults and children (even if no children were included in the current analysis). It seems to me that the EKFC equation performs quite good in most of analyses. It could be a bit more discussed.

Reviewers' comments:

Reviewer's Responses to Questions

**Comments to the Author**

1. If the authors have adequately addressed your comments raised in a previous round of review and you feel that this manuscript is now acceptable for publication, you may indicate that here to bypass the “Comments to the Author” section, enter your conflict of interest statement in the “Confidential to Editor” section, and submit your "Accept" recommendation.

Reviewer #1: All comments have been addressed

Reviewer #2: All comments have been addressed

Reviewer #4: (No Response)

2. Is the manuscript technically sound, and do the data support the conclusions?

Reviewer #1: Yes

Reviewer #2: Yes

Reviewer #4: Yes

3. Has the statistical analysis been performed appropriately and rigorously? 

Reviewer #1: Yes

Reviewer #2: Yes

Reviewer #4: Yes

4. Have the authors made all data underlying the findings in their manuscript fully available?

Reviewer #1: Yes

Reviewer #2: Yes

Reviewer #4: Yes

5. Is the manuscript presented in an intelligible fashion and written in standard English?

Reviewer #1: Yes

Reviewer #2: Yes

Reviewer #4: Yes

6. Review Comments to the Author

Reviewer #1: Thank you for addressing my comments. I would suggest three other minor revisions:

1) in Table 2, serum creatinine should be reported separately for males and females, as the differences between genders are large

2) As the use of a "race coefficient" is highly debated in the US, I would not recommend to calculate yet another "race coefficient" of 1.018 for Black British subjects. As I have said before, the adjustment should be at the creatinine level, not at the GFR-level, which falsely suggests that there are differences in GFR between Black and White people.

3) the authors report "mean" bias in the tables, but this can be largely influenced by outliers (which are always present). In many other studies, "median" bias was the more commonly reported statistic. I would suggest to replace mean bias by median bias (or at least, add median bias).

Reviewer #2: All comments have been well addressed. But, some minors modifications are needed. For example, line 8, page 5 the design of the study is observational cross sectional study (not a cohort study). Line 24, page 18 to refer to the study of Bukabau et al, authors should add in the sentence ''East, Central, West and South Africa report'' that is because the Democratic Republic of the Congo is in Central Africa not in East Africa.

Reviewer #4: The authors have mostly responded to reviewer concerns. One minor point remains:

1. In the first paragraph of the results, please explicitly list "predefined confounders".

7. PLOS authors have the option to publish the peer review history of their article (what does this mean?). If published, this will include your full peer review and any attached files.

Reviewer #1: No

Reviewer #2: **Yes: **Ernest Kiswaya SUMAILI

Reviewer #4: No

---

## [Author Response · Author response to Decision Letter 1]

24 Jul 2021

The article has been largely improved. The authors should be congratulated for that.

Thank you for the kind comments.

I have three minor comments

1) The authors wrote: "This is the largest study exploring the accuracy and impact of eGFR equations in people of African and Afro-Caribbean ancestry outside of Africa or the USA". This is incorrect. Flamant et al (ref 11) studied more European black subjects (n=302) than in the current analysis (n=266). Please tone down.

We apologise for this error, and have modified to the following. ‘This is one of the largest studies’

2) I agree the authors keep their coefficient, even if I also agree with Reviewer that correction should be at the creatinine level, not the GFR level. However, I question the clinical relevance of the correction 1.018. Is it really significant from a clinical perspective? To be discussed.

We agree with this comment and have added that this is unlikely to have clinical importance Page 17 Line 13.

3) It must be reminded that the EKFC equation has the advantage to estimate GFR with the same equation in adults and children (even if no children were included in the current analysis). It seems to me that the EKFC equation performs quite good in most of analyses. It could be a bit more discussed.

Thank you for this suggestion. We have added more detail to the discussion – Page 22 Line 20. 

Reviewer #1: Thank you for addressing my comments. I would suggest three other minor revisions:

1) in Table 2, serum creatinine should be reported separately for males and females, as the differences between genders are large

This detail has now been added to Table 2.

2) As the use of a "race coefficient" is highly debated in the US, I would not recommend to calculate yet another "race coefficient" of 1.018 for Black British subjects. As I have said before, the adjustment should be at the creatinine level, not at the GFR-level, which falsely suggests that there are differences in GFR between Black and White people.

We completely concur with this comment and have amended the discussion to suggest that it should not be used in clinical practice.

3) the authors report "mean" bias in the tables, but this can be largely influenced by outliers (which are always present). In many other studies, "median" bias was the more commonly reported statistic. I would suggest to replace mean bias by median bias (or at least, add median bias).

We have added median bias to the tables.

Reviewer #2: All comments have been well addressed. But, some minors modifications are needed. 

For example, line 8, page 5 the design of the study is observational cross sectional study (not a cohort study). 

Line 24, page 18 to refer to the study of Bukabau et al, authors should add in the sentence ''East, Central, West and South Africa report'' that is because the Democratic Republic of the Congo is in Central Africa not in East Africa. 

Thank you for highlighting these inaccuracies which we have now amended.

Reviewer #4: The authors have mostly responded to reviewer concerns. One minor point remains:

1. In the first paragraph of the results, please explicitly list "predefined confounders".

Thank you for this suggestion. We have now added the predefined confounders again in this section.

---

## [Editor Report · Decision Letter 2]

27 Jul 2021

Estimated Glomerular Filtration Rate Equations in people of self-reported Black ethnicity in the United Kingdom:  Inappropriate adjustment for ethnicity may lead to reduced access to care

PONE-D-20-41065R2

Dear Dr. Bramham,

We’re pleased to inform you that your manuscript has been judged scientifically suitable for publication and will be formally accepted for publication once it meets all outstanding technical requirements.

Kind regards,

Pierre Delanaye

Academic Editor

PLOS ONE
---

## [Editor Report · Acceptance letter]

2 Aug 2021

PONE-D-20-41065R2 

Estimated Glomerular Filtration Rate Equations in people of self-reported Black ethnicity in the United Kingdom :  Inappropriate adjustment for ethnicity may lead to reduced access to care 

Dear Dr. Bramham:

I'm pleased to inform you that your manuscript has been deemed suitable for publication in PLOS ONE. Congratulations! Your manuscript is now with our production department. 

Kind regards, 

on behalf of

Professor Pierre Delanaye 

Academic Editor

PLOS ONE